# Navigating the Pitfalls of Active Learning Evaluation: A Systematic Framework for Meaningful Performance Assessment

**Carsten T. Lüth**[1,2]   **Till J. Bungert**[1,2]   **Lukas Klein**[1,2,3]   **Paul F. Jaeger**[1,2]

[1]German Cancer Research Center (DKFZ) Heidelberg, Interactive Machine Learning Group, Germany
[2]Helmholtz Imaging, German Cancer Research Center (DKFZ) Heidelberg, Heidelberg, Germany
[3]Institute for Machine Learning, ETH Zürich, Switzerland

`carsten.lueth@dkfz-heidelberg.de`

## Abstract

Active Learning (AL) aims to reduce the labeling burden by interactively selecting the most informative samples from a pool of unlabeled data. While there has been extensive research on improving AL query methods in recent years, some studies have questioned the effectiveness of AL compared to emerging paradigms such as semi-supervised (Semi-SL) and self-supervised learning (Self-SL), or a simple optimization of classifier configurations. Thus, today's AL literature presents an inconsistent and contradictory landscape, leaving practitioners uncertain about whether and how to use AL in their tasks. In this work, we make the case that this inconsistency arises from a lack of systematic and realistic evaluation of AL methods. Specifically, we identify five key pitfalls in the current literature that reflect the delicate considerations required for AL evaluation. Further, we present an evaluation framework that overcomes these pitfalls and thus enables meaningful statements about the performance of AL methods. To demonstrate the relevance of our protocol, we present a large-scale empirical study and benchmark for image classification spanning various data sets, query methods, AL settings, and training paradigms. Our findings clarify the inconsistent picture in the literature and enable us to give hands-on recommendations for practitioners. The benchmark is hosted at https://github.com/IML-DKFZ/realistic-al.

## 1   Introduction

Active Learning (AL) is a popular approach to efficiently label large pools of data by interactively querying the most informative samples for training a classification system. While some parts of the AL community actively propose new query methods (QMs) to advance the field [14, 30], others report AL to be generally outperformed by alternative training paradigms to standard training (ST) such as Semi-Supervised Learning (Semi-SL) [20, 43] and Self-Supervised Learning (Self-SL) [6], or even by well-configured standard baselines [44]. To add to the confusion, further studies have shown that AL can even decrease classification performance in certain settings, a phenomenon referred to as the "cold start problem" [6, 20, 43].

This heterogeneous state of research in AL poses a significant challenge for anyone seeking to efficiently annotate their dataset and facing the questions: *On which tasks and datasets is AL beneficial? And how to best employ AL on my dataset?*

37th Conference on Neural Information Processing Systems (NeurIPS 2023).

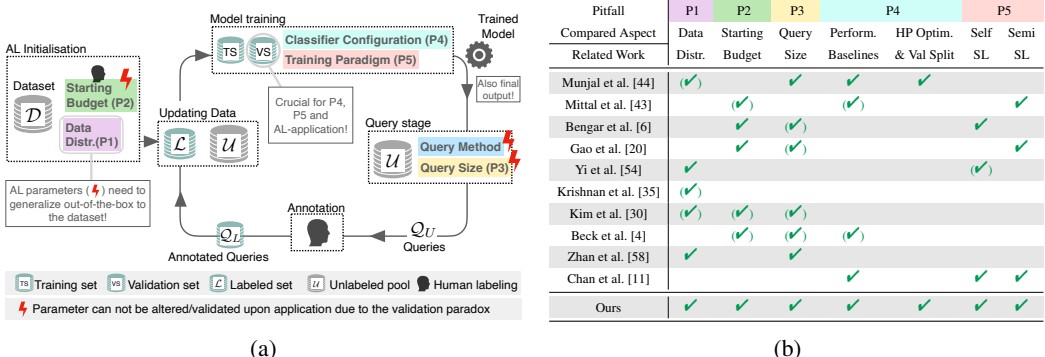

|  | | | | | | |
|---|---|---|---|---|---|---|
| **(a)** | | | | **(b)** | | |

| Pitfall | P1 | P2 | P3 | P4 | | P5 | |
|---|---|---|---|---|---|---|---|
| Compared Aspect | Data | Starting | Query | Perform. | HP Optim. | Self | Semi |
| Related Work | Distr. | Budget | Size | Baselines | & Val Split | SL | SL |
| Munjal et al. [44] | (✔) | | ✔ | ✔ | ✔ | | |
| Mittal et al. [43] | | (✔) | | (✔) | | | ✔ |
| Bengar et al. [6] | | ✔ | (✔) | | | ✔ | |
| Gao et al. [20] | | ✔ | (✔) | | | | ✔ |
| Yi et al. [54] | ✔ | | | | (✔) | | |
| Krishnan et al. [35] | (✔) | | | | | | |
| Kim et al. [30] | (✔) | (✔) | (✔) | | | | |
| Beck et al. [4] | | (✔) | (✔) | (✔) | | | |
| Zhan et al. [58] | ✔ | | ✔ | | | | |
| Chan et al. [11] | | | | ✔ | | ✔ | ✔ |
| Ours | ✔ | ✔ | ✔ | ✔ | ✔ | ✔ | ✔ |

Figure 1: **(a)**: The five pitfalls (P1-P5) for meaningful evaluation in the context of the Active Learning loop. Detailed information is provided in Sec. 2. **(b)**: The five pitfalls are highly prevalent in the current literature (green ticks denote successful avoidance of the respective pitfall). A detailed correspondance between individual studies and pitfalls is provided in Appendix B. Our study is the first to avoid all pitfalls and enable trustworthy performance assessment of AL methods.

We argue that the inconsistency arises from a lack of systematic and realistic evaluation of AL methods. AL inherently comes with specific requirements on how methods will be applied in practice. First and foremost, the stated purpose of "reducing the labeling effort on a task" implies that AL methods need to be rolled out to unseen datasets different from the labeled development data. This inherent requirement of cross-task generalization needs to be reflected in method evaluation, posing a need to test methods under diverse settings to identify robust and trustworthy configurations for the subsequent "blind" application on real-life tasks (see "validation paradox", Sec. 2.2). However, such considerations are generally neglected in AL research, as identified in our work by means of five key pitfalls in the current literature, spanning from a lack of tested AL settings and tasks to a lack of appropriate baselines (see Figure 1 and P1-P5 in Sec. 2.3).

To this end, we present an evaluation framework for deep active classification that overcomes the five pitfalls and demonstrate the relevance of this contribution by means of a large-scale empirical study spanning various datasets, QMs, AL settings, and training paradigms. To give one concrete example for the disruptive impact of our work: We address the widespread pitfall of neglecting classifier configuration (see P4 in Figure 1b) by introducing a light-weight protocol for high-quality configuration. The results demonstrate how this simple protocol lets even our random-query baseline exceed the originally reported performance of recent AL methods [1, 30, 35] on the widely-used CIFAR-10/100 datasets, while drastically reducing computational effort compared to the configuration protocol of Munjal et al. [44].

This example showcases, that the novelty of our work does not lie in presenting entirely novel results and insights, but in the fact that our comprehensive and systematic approach is the first to address all five key pitfalls and thus to provide *trustworthy* insights into the real-life capabilities of current AL methods. By relating these insights to recent studies that have been subject to flawed evaluation, we are able to resolve existing inconsistencies in the field and provide robust guidelines for when and how to apply AL on a given task.

## 2 Realistic Evaluation in Active Learning

### 2.1 Active Learning Task Formulation

As depicted in Figure 1a, AL describes a classification task, where a dataset $\mathcal{D}$ is given that is divided into a labeled set $\mathcal{L}$ and an unlabeled pool $\mathcal{U}$. Initially, only a fraction of the data is labelled ("starting budget"), which is ideally split further into a training set and a validation set for hyperparameter (HP) tuning and performance monitoring. After initial training, the QM is used to generate queries $\mathcal{Q}_{\mathcal{U}}$ of a certain amount ("query size") that represent the most informative samples from $\mathcal{U}$ based on the current classifier predictions. Subsequently, queried samples are labeled ($\mathcal{Q}_{\mathcal{L}}$), moved from $\mathcal{U}$ to $\mathcal{L}$, and the classifier is re-trained on $\mathcal{L}$. This process is repeated until classifier performance is satisfying, as monitored on the validation set.

## 2.2 Overview over critical concepts in Active Learning evaluation

Evaluating an AL algorithm typically means testing how much classification performance is gained by data samples queried over several training iterations. The QM selecting those samples is considered useful if the performance gains exceed the gains of randomly queried samples. While this process is well-established, it is prone to neglecting critical concepts for evaluation and thus to over-simplification of how AL algorithms are applied in practice.

**AL validation paradox.** When applying AL on a new, mostly unlabelled dataset, one can not directly validate which QM type or configuration performs best, as this would require labeling individual AL trajectories through the dataset for each setting. This excessive labeling would directly contradict the goal of AL to reduce the labeling effort, a predicament which we refer to as the AL validation paradox. Notably, this is in contrast to standard ML, where fixed labeled training and validation splits generally suffices for model selection and configuration.

**Special requirements on evaluation.** As the described paradox impedes on-the-spot validation, it forces one to, instead, estimate how well certain QMs will perform on the given task based solely on prior knowledge. The quality of this prior knowledge depends on how extensively the respective QM has been validated prior to application, i.e. on the development data. This implies several critical requirements on evaluation:

- A generalization gap between development and application arises, because the latter typically comes with practical constraints such as on the size of the starting budget, the query size, or a given class imbalance. To avoid AL failure, such constraints need to be anticipated by *evaluating QMs on a realistic range of the corresponding design choices*.

- Given these inherent generalization requirements in AL, it is crucial to simulate the generalization process by evaluating QMs on *roll-out datasets*, i.e. tasks that were not part of the configuration process on the *development datasets*.

- Evaluation needs to acknowledge that AL is an elaborate and costly training paradigm, and thus ensure that methods are *tested against simpler and cheaper alternatives*. For instance, if the gain of a QM over random sampling vanishes by simply tuning the learning rate of the classifier, or by a single self-supervised pretraining, the QM, and AL in general, provides no practical value.

To effectively describe the current oversight of these requirements in the field, we identified five concrete evaluation pitfalls (P1-P5), which are detailed in Section 2.3.

**Cold start problem.** Neglecting these pitfalls could render AL ineffective, or even counterproductive, for specific tasks, as it might result in QMs underperforming relative to a random-sampling baseline. Such failures occur predominantly in few-label settings and are a recognized challenge in the research community, often termed as the cold start problem [20].

## 2.3 Current pitfalls of Active Learning evaluation

The current AL literature features an inconsistent landscape of evaluation protocols, but, as Figure 1b shows, none of them adhere to all requirements for evaluation described above. To study the current oversight of these requirements, we identify five key *pitfalls* (P1-P5) that need to be overcome for meaningful evaluation of QMs. For a visual overview of the pitfalls and how they integrate into the AL setting see Figure 1a.

**P1: Lack of evaluated data distribution settings.** To ensure that QMs work out-of-the-box in real-life settings, they need to be evaluated on a broad data distribution. Relevant aspects of a distribution in the AL-context go beyond the data domain and include class distribution, the relative difficulty of separation across classes, as well as a potential mismatch between the frequency and importance of classes. All of these aspects directly affect the functionality of a QM and may lead to real-life failure of AL when not considered in the evaluation.

*Current practice:* Most current work is limited to evaluating QMs on balanced datasets from one specific domain (e.g. CIFAR-10/100) and under the assumption of equal class importance. To our knowledge, testing the generalizability of a fixed QM setting to new datasets ("roll-out") has not been performed before. There are some experiments conducted on an artificially imbalanced dataset

(CIFAR-LT) [35, 44] suggesting good AL performance. Further, Gal et al. [19] study AL on the ISIC-2016 dataset, but obtain volatile results due to the small dataset size. Atighehchian et al. [3] study AL on the MIO-TCD dataset and reported performance improvements for underrepresented classes. In the large study Beck et al. [4] perform experiments on five mostly balanced datasets datasets and [58] use 13 datasets for mostly standard AL experiments.

$\rightarrow$ *Proposed solution:* We argue that the underrepresentation of class-imbalanced datasets in the field is one reason for current doubts regarding the general functionality of AL. Real-life settings will most likely not be class balanced providing a natural advantage of AL over random sampling. We propose to consider diverse datasets with real class imbalances as an essential part of AL evaluation and advocate for the inclusion of "roll-out" datasets, as a real-life test of selected and fixed AL settings.

**P2: Lack of evaluated starting budgets.** There are two reasons for why this parameter is an essential aspect of AL evaluation: 1) Upon application, the budget might be fixed and the QM is required to generalize out-of-the-box to this setting. 2) We are interested in the minimal budget at which the QM works since a too large budget implies inefficient labeling (equivalent to random queries) and a too small budget is likely to cause AL failure (cold start problem). This search needs to be performed prior to an AL application due to the validation paradox.

*Current practice:* Most recent studies evaluate AL on a single starting budget made of thousands of samples on datasets such as CIFAR-10 and CIFAR-100[30, 44, 54, 55]. Information-theoretic publications commonly use a smaller starting budget [19, 32], but typically on even simpler datasets such as MNIST [38]. Beck et al. [4] compare two starting budgets on MNIST reporting no performance drop. On the other hand, some studies benchmarking AL against Semi-SL and Self-SL compare two [6] or three [20, 43] starting budgets often with the conclusion that smaller starting budgets lead to AL failure. Bengar et al. [6] report that there exists a relationship between the number of classes in a task and the optimal starting budget (the intuition being that class number is a proxy for task complexity).

$\rightarrow$ *Proposed solution:* To overcome this pitfall and resolve the current contradictions, we evaluate all QMs for three different starting budgets on all datasets. We refer to these settings as the *low-, mid-, and high-label regime*. Extending on the findings of Bengar et al. [6], adequate budget sizes are determined using heuristics based on the number of classes per task.

**P3: Lack of evaluated query sizes.** The number of samples queried for labelling in each AL iteration is an essential aspect of QM evaluation. This is because, upon application, this parameter might be predefined by the compute-versus-label cost ratio of the respective task (a smaller query size amounts to higher computational efforts but might enable more informed queries and thus less labeling). Since query size cannot be validated on the task at hand due to the validation paradox, the generalizability of QMs to various settings of this parameter needs to be evaluated beforehand.

*Current practice:* In current literature, there is a concerning disconnect between theoretical and practical papers regarding what constitutes a reasonable query size. Information-theoretical papers typically select the smallest query size possible and QMs such as BatchBALD [32] are specifically designed to simulate reduced query sizes [19, 46]. In contrast, practically-oriented papers usually employ larger query sizes [30, 43, 44, 51, 55], but only in combination with large starting budgets (P2), where cold start problems generally do not occur. Only a few studies perform limited evaluations of varying query sizes. Beck et al. [4] and Zhan et al. [58] report a negligible effect of varying query sizes, but only evaluate in combination with large starting budgets (1000 samples). In line with this, Munjal et al. [44] conclude that the choice of query size does not matter, but only compared two large values (2500 versus 5000 samples) on a fixed large starting budget (5000 samples). Atighehchian et al. [3] come to a similar conclusion, but also only considered a relatively large starting budget (500) for ImageNet-pretrained models on CIFAR-10, where, again, no cold start problem occurs. Bengar et al. [6] employ varying query sizes without further analysis of the parameter.

$\rightarrow$ *Proposed solution:* To overcome this pitfall and reliably study the effect of query sizes also in low-label, i.e. high-risk, settings, we evaluate all QMs for three different query sizes in combination with varying starting budgets on all datasets (i.e. as part of the low-, mid-, and high-label regimes). For a specific focus on the effect of query size in the low-label settings, we perform an additional

ablation with varying query sizes on a small fixed starting budget.

**P4: Neglection of the classifier configuration.** As stated in Sec. 2.2, when aiming to draw conclusions about the performance or usefulness of a QM, it is critical that this evaluation be based on well-configured classifiers. Otherwise, performance gains might be attributed to AL that could have been achieved by simple hyperparameter (HP) modifications. Separating a validation split from the training data is a crucial requirement for sound HP tuning.

*Current practice:* Most studies in AL literature do not report how HPs are obtained and do not mention the use of validation splits [30, 35, 43, 51, 55]. Typically, reported settings are copied from fully labeled data scenarios. In some cases, even the proposed QMs feature delicate HPs without reporting how they were optimized raising the question of whether these settings generalize to new data [30, 51, 55]. Munjal et al. [44] demonstrate how adequate HP tuning on a validation set allows a random query baseline to outperform current QMs under their originally proposed HP settings. However, they run a full grid search for every QM and AL training iteration, which might not be feasible in practice.

$\rightarrow$ *Proposed solution:* To overcome this pitfall and enable meaningful performance assessment of AL methods, we define a validation dataset of a size deducted heuristically from the starting budget. Based on this data, a small selection of HPs (learning rate, weight decay and data augmentations [17]) is tuned only once per AL experiment while training on the starting budget. The limited search space and discarding of multiple tuning iterations result in a lightweight and practically feasible protocol for classifier configuration.

**P5: Neglection of alternative training paradigms.** Analogously to arguments made in P4, meaningful evaluation of AL requires comparison against alternative approaches that address the same problem. Specifically, the training paradigms Self-SL [13, 25] and Semi-SL [39, 52] have shown strong potential to make efficient use of an unlabeled data pool in a classification task thus alleviating the labeling burden. Additionally to benchmarking AL against Self-SL and Semi-SL, the question arises of whether AL can yield performance gains when combined with these paradigms.

*Current practice:* While most AL studies do not consider Self-SL and Semi-SL, there are a few recent exceptions: Bengar et al. [6] benchmark AL in combination with Self-SL and conclude that AL only yields gains under sufficiently high starting budgets. However, these results suffer from inadequate classifier configuration (P4). Yi et al. [54] propose a QM in combination with Self-SL, but the employed Self-SL strategy is limited by compatibility with the proposed QM. Further, Gao et al. [20] combine Semi-SL with AL and report superior performance compared to ST for CIFAR-10/100 and ImageNet [48], i.e. the datasets on which Semi-SL methods have been developed. Similarly, Mittal et al. [43] evaluate the combination of Semi-SL and AL on CIFAR-10/100, reporting strong improvements compared to ST and find that AL decreases performance for small starting budgets.

$\rightarrow$ *Proposed solution:* To overcome this pitfall and resolve current inconsistencies, we benchmark all QMs against both Self-SL and Semi-SL, and evaluate the combination of AL with these paradigms. Crucially, we are the first to study these relations as part of a reliable evaluation, i.e. while avoiding all other key pitfalls (P1-P4).

## 3 Experimental Setup

This section describes the design of our empirical study in light of the proposed improvements for AL evaluation (detailed experimental settings can be found in Appendix D). We first address P1 by extending our evaluation to 5 different datasets, containing different label distributions. Specifically, these datasets include CIFAR-10, CIFAR-100, CIFAR-10 LT, ISIC-2019 and MIO-TCD, where the first three are developmental datasets and the latter two are used exclusively for the proposed roll-out evaluation. Further, we address P2 and P3 by defining three different *label regimes* which we refer to as "low-label", "mid-label" and "high-label" regimes. Starting budgets and query sizes are both set to $5 \times C$, $25 \times C$ and $100 \times C$ for the three label regimes, where $C$ denotes the number of classes. [1] To address P4, we configure our classifiers for all three label regimes based on a validation set five

---

[1] These deviate for CIFAR-100 [$5 \times C$ (low-), $10 \times C$ (mid-), $50 \times C$ (high-label)] due to a smaller ratio of dataset size to number of classes.

times the size of the starting budget. Further, addressing P4 and P5 we use a ResNet-18 [24] as the backbone in all experiments and optimize the essential HPs for each respective training paradigm. At last, we address P5 by comparing randomly initialized models (standard training) against Self-SL pre-trained models and Semi-SL models.

**Compared query methods.** In this section, we describe the applied QM in more general terms and refer to Appendix A for details. We focus exclusively on QMs which do not alter the classifier and require no configuration of additional HPs except for bayesian QMs which are based on dropout. Generally, QMs can be divided into two categories as they are either based on uncertainty estimation or enforcing exploration. **Random:** The baseline all QMs are compared against which randomly draws samples from the pool $\mathcal{U}$. **Core-Set:** This explorative QM aims to find the core-set of a convolutional neural network [49] by means of a K-Center Greedy approximation on intermediate representations. **Entropy:** This uncertainty-based QM selects the samples with the highest entropy across predicted class scores [50]. **BALD:** This uncertainty-based QM uses the mutual information between the class label and the model parameters with regard to each sample for greedy selection [27], it was introduced with dropout for deep bayesian active learning[19]. **BADGE:** This QM performs a clustering based on per-sample gradient vectors obtained via proxy labels. This enables a selection that is both diverse and guided by uncertainty [2].

**Datasets.** The initial datasets for our experiments are **CIFAR-10/100** [36]. For further analysis we added **CIFAR-10 LT** [10], an artificially created dataset built upon CIFAR-10 with a long-tail distribution of classes following an exponential decay for the training split. The imbalance factor $\rho$ was selected to be 50, following [35]. On all three of these datasets, we use accuracy on the class-balanced test set as the primary performance metric. Finally, we selected **ISIC-2019**[15, 16, 53] and **MIO-TCD**[42] as roll-out datasets to verify the generalizability of AL methods to new settings. Both of these datasets feature inherently imbalanced class distributions, and are more likely to be subject to label noise compared to the CIFAR datasets. As such, we deem the two roll-out datasets an essential step toward a realistic evaluation of AL methods. For both MIO-TCD and ISIC-2019, we use balanced accuracy as the primary performance measure (Appendix D).

**Active learning setup.** We report performance measures for each dataset on identical test splits based on three experiments using different seeded models and different train and validation splits to reduce the possible influence of these parameters on our results. Further, we train the models from scratch on every training step to avoid correlated queries [32].

**Training paradigms.** Randomly initialized and supervised-trained models are referred to as ST models. Further, we use the popular contrastive SimCLR [13] training as a basis for Self-SL pre-training. These models are fine-tuned and are referred to as Self-SL models. Self-SL models have a two-layer MLP as a classification head to make better use of the representations (ablation in Appendix E). For ST and Self-SL models, we obtain bayesian models by adding dropout to the final representations before the classification head following [19]. As a Semi-SL method, we use FixMatch [52] which combines the principles of consistency and uncertainty reduction. Due to the long training times (factor 80) compared to ST training, we only ran experiments in the low- and mid-label regime while increasing the query size by a factor of three to reduce training costs.

**Hyperparameter selection.** The HPs for our models are selected for each label regime before the AL loop is started using the corresponding validation set. For Self-SL and ST models we use a fixed training recipe (Scheduler, Optimizer, Batch Size, Epochs) and only optimize learning rate, weight decay and data augmentations. The data augmentations used are standard augmentations and Randaugment which uses stronger augmentations acting as a regularization [17]. For Semi-SL methods we fix all HPs with the exception of learning rate and weight decay following Sohn et al. [52]. Model selection for ST and Self-SL models is based on the best validation set epoch, while for Semi-SL models the final checkpoint is used. For imbalanced datasets, we use oversampling for ST and Self-SL pre-trained models Buda et al. [8] and use weighted cross-entropy-loss and distribution alignment for FixMatch.

**Low-Label query size ablation.** To investigate the effect of query size in the low-label regime, we conduct an ablation with Self-SL pre-trained models on CIFAR-100 and ISIC-2019. For CIFAR-100 the query sizes are 50, 500 and 2000, while for ISIC-2019 they are 10, 40 and 160.

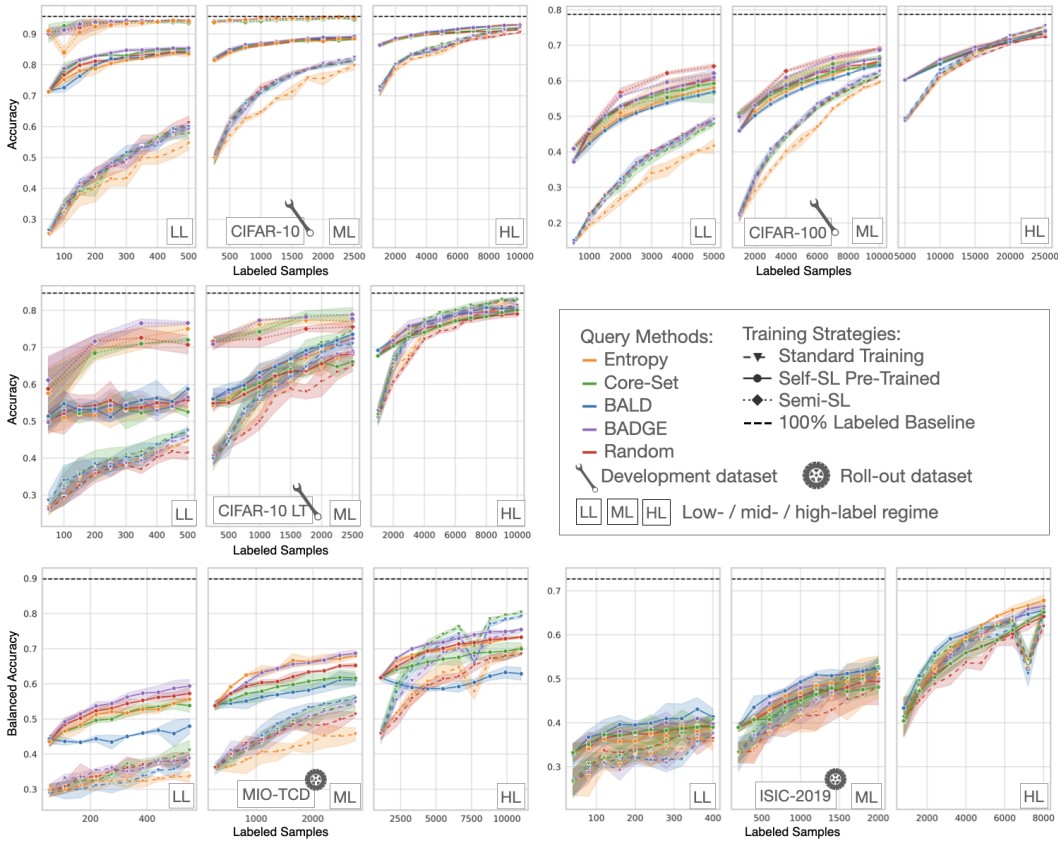

Figure 2: Results obtained with our proposed evaluation protocol over five different datasets and the three label regimes. These experiments are to our knowledge the largest conducted study for AL and reveal insights along the lines of the five key parameters as discussed in Sec. 2. The strong performance dip on MIO-TCD and ISIC-2019 is discussed in Sec. 4.

## 4    Results & Discussion

The results of our empirical study are shown in Figure 2. An in-depth analysis of results for individual datasets and analysis based on the pairwise penalty matrix [2] and area under the budget curve [57] can be found in Appendix F. Here, we will discuss the main findings along the lines of our five identified pitfalls of evaluation (P1-P5). Our findings demonstrate the relevance of the proposed protocol for realistic evaluation and its potential to generate trustworthy insights on when and how AL works. If not otherwise mentioned, all references in this chapter refer to AL studies.

**P1 Data distribution.** The proposed evaluation over a diverse selection of dataset distributions including specific roll-out datasets proved essential for realistic evaluation of QMs as well as the different training strategies. One main insight is the fact that class distribution is a crucial predictor for the potential performance gains of AL on a dataset: Performance gains of AL are generally higher on imbalanced datasets and occur consistently even for ST models with a small starting budget, which are typically prone to experience cold start problems. This observation is consistent with a few previous studies [30, 35, 54]. Further, our results underpin the importance of the roll-out datasets e.g. when looking at the sub-random performance of BALD (with Self-SL) and Entropy (with ST) on MIO-TCD. Such worst-case failures of AL application (increased compute and labeling effort due to AL) could not have been predicted based on development data where all AL-parameters are optimized. Another example is the lack of generalizability of Semi-SL, where performance in relation to other Self-SL and ST decreases gradually with data complexity (going from CIFAR-10 to CIFAR-100/-10 LT to the two roll-out datasets MIO-TCD and ISIC-2019.)

**P2 Starting budget.** The comprehensive study of various starting budgets on all datasets reveals that AL methods are more robust with regard to small starting budgets than previously reported

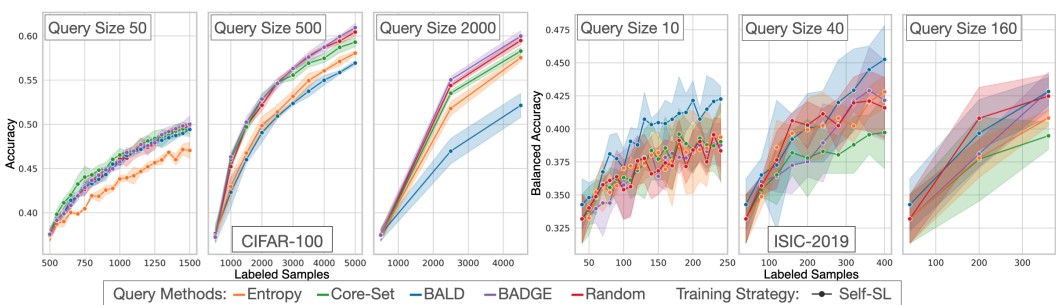

Figure 3: Low-label query size ablation on CIFAR-100 and ISIC-2019. On CIFAR-100, reducing the query size resolves the observed failure mode of BALD. However, no improvement is observed on ISIC-2019, presumably because BALD already shows the best performance. Further, BADGE performs consistently well across all query sizes without failure modes, revealing its robustness also for low-label settings.

[6, 20, 43]. With the exception of Entropy we did not observe cold start problems even for any QM even in combination with notoriously prone ST models. The described robustness is presumably enabled by our thorough classifier configuration (P4) and heuristically adapted query sizes (P3). This finding has great impact potential suggesting that AL can be applied at earlier points in the annotation process thereby further reducing the labeling cost, especially in combination with BADGE which performed consistently better or on par with Random. Similarly for Self-SL models, AL performs well on small starting budgets with the exceptions of CIFAR-100 (BALD, Entropy) and MIO-TCD (BALD, Core-Set, Entropy).

**P3 Query size.** Based on our evaluation of the query size we can empirically confirm its importance with regard to 1) general AL performance and 2) counteracting the cold start problem. The are, however, surprising findings indicating that the exact interaction between query size and performance remains an open research question. For instance, we observe the cold start problem for BALD with Self-SL on CIFAR-100 ($\sim 50\%$ accuracy at 2k labeled samples for query sizes of 500 (low-label regime) and 1k (mid-label regime). On the other hand, in the high label regime (budget of 5k and query size of 5k) ST and Self-SL models with similar accuracies of 50% and 60%, respectively, benefit from BALD. Since cold start problems are commonly associated with large query sizes, this finding seems counter-intuitive, but has been reported before although without further investigations [6, 20, 43]. To gain a better understanding of this phenomenon and how QMs interact with query size in the low-label regime, we performed a dedicated experiment series for Self-SL training on CIFAR-100 and ISIC-2019 (see Figure 3 and Appendix F.2). Due to a significant improvement of BALD on CIFAR-100 for even smaller query sizes from sub-Random to Random performance and no worsening on ISIC-2019, we can conclude that small query sizes represent an effective countermeasure against the cold-start problem for BALD contrary to the findings of [44, 58] and currently not considered as a solution [6, 20, 43]. This potentially explains the gap between these findings and more theoretical works which advertise using the smallest query size possible [19, 33]. Importantly, while smaller query sizes appear as a potential solution for the observed instabilities of BALD, they seem to have no considerable effect on the other QMs. Moreover, our ablation reveals that even in the low-label settings, BADGE is the overall most reliable of all compared QMs and exhibits no sub-Random performance. This adds to the existing reports of the robustness of BADGE for higher label regimes [4, 58].

**P4 Classifer configuration.** Our results show that method configuration on a properly sized validation set is essential for realistic evaluation in AL. For instance, our method configuration has the same effect on the classifier performance as increasing the number of labeled training samples by a factor of $\sim 5$ (e.g. our ST model reach approximately the same accuracy of $\sim 44\%$ trained on 200 samples compared to models in the AL study by Bengar et al. [6] trained on 1k samples). The effectiveness of our proposed lightweight HP selection on the starting budget including only three parameters (Sec. 3) is demonstrated by the fact that all our ST models substantially outperform respective models found in relevant literature [6, 20, 30, 35, 43, 54, 55] where HP optimization is generally neglected. Details for this comparison are provided in Appendix H. This raises the question of to which extent reported AL advantages could have been achieved by simple classifier configurations. Further, our models

also generally outperform expensively configured models by Munjal et al. [44]. Thus, we conclude that manually constraining the search space renders HP optimization feasible in practice without decreasing performance and ensures performance gains by Active Learning are not overstated. The importance of the proposed strategy to optimize HPs on the starting budget for each new dataset is supported by the fact that the resulting configurations change across datasets.

**P5 Alternative training paradigms.** Based on our study benchmarking AL in the context of both Self-SL and Semi-SL, we see that while Self-SL generally leads to improvements across all experiments, Semi-SL only leads to considerably improved performance on the simpler datasets CIFAR-10/100, on which Semi-SL methods are typically developed. Generally, models trained with either of the two training paradigms receive a lower performance gain from AL (over random querying) compared to ST. Crucially, Self-SL models converge around $2.5$ times faster than ST models, while the training time of Semi-SL models is around $80$ times longer than ST and often yields only small benefits over Self-SL or ST models. The fact that AL entails multiple training iterations amplifies the computational burden of Semi-SL, rendering their combination prohibitively expensive in most practical scenarios. Further, the fact that our Semi-SL models based on Fixmatch do not seem to generalize to more complex datasets in our setting stands in stark contrast to conclusions drawn by [20, 43] as to which the emergence of Semi-SL renders AL redundant. Interestingly, the exact settings where Semi-SL does not provide benefits in our study are the ones where AL proved advantageous. The described contradiction with the literature underlines the importance of our proposed protocol testing for a method's generalizability to unseen datasets. This is especially critical for Semi-SL, which is known for unstable performance on noisy and class imbalanced datasets as noted by studies focusing on Semi-SL [7, 22, 23, 29, 45, 56].

**Limitations.** We propose a light-weight and practically feasible strategy for HP optimization and made other design choices (e.g. ResNet-18 classifier), thus we can not guarantee that our configurations are optimal for all compared training paradigms and would like to provide a critical discussion: **1)** The ResNet-18 in combination with our shortened training times, might hinder Semi-SL performance more than other training paradigms. This setting is necessary to be able to cope with the computational cost of Semi-SL (factor $\sim 400$ training time compared with ST). **2)** The validation set size of $5\times$ starting budget size (i.e. training set) could be considered as larger than practically desirable, where most data would be used for training. This design decision follows the study of Oliver et al. [45], showing that an adequately sized validation set is necessary for proper HP selection (especially for Semi-SL). **3)** We observe a performance dip of ST models on MIO-TCD and ISIC-2019 at $\sim$7k samples, which we attribute to our HP selection scheme. This indicates that HPs might need to be re-selected occasionally at certain training iterations. However, such cases are immediately detected in practice allowing for correction where necessary by re-optimizing the HPs (see Appendix I.1). A more extensive discussion of limitations can be found in Appendix I.

## 5   Conclusion & Take-Aways

Our experiments provide strong empirical evidence that the current evaluation protocols do not sufficiently answer the key question every potential AL practitioner faces: Should I employ AL on my dataset? Answering this question entails estimating whether an AL algorithm will provide performance gains over random queries and thus whether the expected reduction in labeling cost outweighs both the additional computational and engineering cost attributed to AL. We argue that our proposed protocol for realistic evaluation represents a cornerstone towards enabling informed decisions in this context. This is made possible by focusing on evaluating the generalizability of AL to new settings under real-world conditions. This perspective manifests itself in the form of five key pitfalls in the current AL literature regarding data distribution, starting budget, query size, classifier configuration, and alternative training paradigms (see Sec. 2.3). Even though the thorough evaluation increases the computational cost once during method development, it will lead to a net cost reduction in the long-term by enabling an informed selection of AL methods in practice, thereby effectively reducing annotation cost upon application.

We hope that the proposed evaluation protocol in combination with the publicly-available benchmark will help to push active learning towards robust and wide-spread real-world application.

**Main empirical insights revealed by our study:**

- Assessment of AL methods in the literature is substantially hampered by subpar classifier configurations, but meaningful assessment can be restored by our protocol for lightweight hyperparameter tuning on the starting budget.
- AL generally provides substantial gains in class-imbalanced settings.
- BADGE is the best-performing QM across a realistic range of datasets, starting budgets, and query sizes and exhibits no failure modes (i.e. sub-Random performance).
- Combining AL with Self-SL considerably improves performance, shortens the training time, and stabilizes optimization, especially in low-label settings.
- FixMatch results indicate that Semi-SL methods perform well on datasets where they have been developed, but may struggle to generalize to more realistic scenarios such as class imbalanced data. Combining the paradigm with AL additionally suffers from extensive training times.
- BALD with Self-SL pre-trained models benefits from smaller query sizes on small starting budgets possibly circumventing the "cold start problem".

**Take-aways for developing and proposing new Active Learning algorithms:**

AL methods should be tested for generalizability on roll-out datasets and come with a clear recipe for real-world application including how to adapt all design choices to new settings. Since the expected benefit of AL on a new setting increases with lower application costs, we believe there is a high potential for wide-spread real-world use of AL by reducing the two prevalent cost factors: **1)** Engineering costs, which are reduced by building easy-to-use AL tools. **2)** Computational costs, which are reduced by explicitly including methods that shorten the training time in AL.

**Take-aways for the cost-benefit analysis of deploying Active Learning:**

1. Since we identify BADGE as a robust query method exhibiting no sub-random-sampling performance across all tested settings, the potential harm of AL is minimized allowing to base decisions around deploying AL on the described cost-benefit analysis.

2. The expected benefit is high in settings, where there is a mismatch between the task-specific importance of individual classes and their frequency in the dataset (e.g. class-imbalanced datasets in combination with tasks requiring a balanced classifier).

3. The expected benefit further increases with the labeling cost that will be reduced by AL. AL is thus likely to yield a net benefit in settings with high label cost and low computational and engineering costs.

**Future research:**

Future research may investigate to what extend AL benefits from foundation models such as CLIP [47], which have been shown to reach strong performance in low-label settings through finetuning or knowledge extraction. Additionally, their rapid re-finetuning during AL iterations may enable real-time interactive labelling. A further important aspect might be to study the effect of AL training on the classifiers' ability to detect outliers [18] or catch failures under distributions shifts [28].

## Acknowledgements

This work was funded by Helmholtz Imaging (HI), a platform of the Helmholtz Incubator on Information and Data Science. This work is supported by the Helmholtz Association Initiative and Networking Fund under the Helmholtz AI platform grant (ALEGRA (ZT-I-PF-5-121)). We thank Maximilian Zenk, Sebastian Ziegler and Fabian Isensee for insightful discussions and feedback.

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

# A  Active learning, in more detail

First, we will give an additional task description of Active Learning (AL) in Appendix A.1 introducing necessary concepts and mathematical notation for our compared query methods which are discussed in Appendix A.2. Finally, we give intuitions where the connection of AL to Self-Supervised Learning (Self-SL) and Semi-Supervised Learning (Semi-SL) lies in Appendix A.3 and Appendix F.4.

## A.1  From supervised to active learning

In supervised learning, we are given a labeled dataset of sample-target pairs $(x, y) \in \mathcal{L}$ sampled from an unknown joint distribution $p(x, y)$. Our goal is to produce a prediction function $p(Y|x, \theta)$ parametrized by $\theta$, which outputs a target value distribution for previously unseen samples from $p(x)$. Choosing $\theta$ might amount, for example, to optimizing a loss function which reflects the extent to which $\mathrm{argmax}_c p(Y = c|x, \theta) = y$ for $(x, y) \in \mathcal{L}$. However, in pool-based AL we are additionally given a collection of unlabeled samples $\mathcal{U}$, sampled from $p(x, y)$. By using a query method (QM), we hope to leverage this data efficiently through querying and successively labeling the most informative samples $(x_1, ..., x_B)$ from the pool. This should lead to a prediction function better reflecting $p(Y|x)$ than if random samples were queried.

From a more abstract perspective, the goal of AL is to use the information of the labeled dataset $\mathcal{L}$ and the prediction function $p(Y|x, \theta)$, to find the samples giving the most information where $p(Y|x)$ deviates from $p(Y|x, \theta)$. This is also reflected by the way performance in AL is measured – which is the relative performance gain of the prediction function with queries from a QM compared to random queries. Making the prediction function implicitly the gauge for measuring the "success" of an AL strategy.

At the heart of AL is an optimization problem: AL is a game of reducing cost – one trades in computation cost with the expectation of lowering labeling cost which is deemed to be the bottleneck.

## A.2  Query methods

A comprehensive overview of AL and its methods is out of the scope of this paper, we refer interested readers to [50] as a basis and [40] for an overview of current research. Most QMs fall into two categories following either: explorative strategies, which enforce queried samples to explore the distribution of $p(x)$; and uncertainty-based strategies, which make direct use of the prediction function $p(Y|x, \theta)$. [2] The principled combination of both strategies, especially to allow larger query sizes for uncertainty-based QM, is an open research question.

In our work, we focus exclusively on QMs which induce no changes to the prediction function and add no additional HPs except for Bayesian QMs modeled with dropout. This immediately rules out QMs like Learning-Loss [55] or TA-VAAL [30], changing the prediction function, and QMs like VAAL [51], introducing new HPs. The QMs we use for our comparisons are currently state-of-the-art in AL on classification tasks. Further, they require no additional hyperparameters (HPs) to be set for the query function which is hard to evaluate in practice due to the validation paradox.

For this chapter we follow the notation introduced in [31], where e.g. $X$ represents a random variable and $x$ represents a concrete sample of variable $X$.

**Random**  Draws samples from the pool $\mathcal{U}$ randomly which follows $p(x, y)$. Therefore it can be interpreted as an exploratory QM.

**Core-Set**  This method is based on the finding that the decision boundaries of convolutional neural networks are based on a small set of samples. To find these samples, the Core-Set QM queries samples that minimize the maximal shortest path from unlabeled to labeled sample in the representation space of the classifier [49]. This is also known as the K-Center problem, for which we use the K-Center greedy approximation. It draws queries especially from tails of the data distribution,

---

[2]This is conceptually similar to the exploration and exploitation paradigm seen in Reinforcement Learning and there actually exist strong parallels between Reinforcement Learning and Active Learning – so much so that Reinforcement Learning has been proposed to use in AL and AL-based strategies have been proposed to be used in Reinforcement Learning.

to cover the whole dataset as well as possible. Therefore we classify it as an explorative strategy.

**Entropy**   The Entropy QM greedily queries the samples $x$ with the highest uncertainty of the model as shown in Equation (1) with $C$ being the number of classes.

$$H(Y|x,\theta) = \sum_{c=1}^{C} p(Y=c|x,\theta) \cdot \log(p(Y=c|x,\theta)) \tag{1}$$

**BALD**   Uses a bayesian model and selects greedily a query of samples with the highest mutual information between the predicted labels $Y$ and weights $\Theta$ for a sample $x$ following [19]. From the weight variable $\Theta$ the concrete values $\theta \sim p(\theta|\mathcal{L})$ are then obtained by MC sampling of a bayesian dropout model [19].

$$\begin{aligned}
\text{MI}(Y;\Theta|x,\mathcal{L}) = \sum_{c=1}^{C} &(p(Y=c|x,\mathcal{L}) \cdot \log(p(Y=c|x,\mathcal{L})) \\
&- \mathbb{E}_{p(\theta|\mathcal{L})}\left[H(Y|x,\theta)\right])
\end{aligned} \tag{2}$$

Where $p(Y|x,\mathcal{L}) = \mathbb{E}_{p(\theta|\mathcal{L})}\left[p(Y|x,\theta)\right]$.

**BADGE**   Uses the K-MEANS++ initialization algorithm on the last layer gradient embeddings $\mathcal{G} = \{\nabla_{\theta_{L-1}}\mathcal{L}(\hat{y}(x), p(Y|x,\theta)|x \in \mathcal{U}\}$ to obtain B centers. These are likely to have diverse directions (which capture diversity due to diverse parameter updates) and large loss gradients (capturing uncertainty due to large loss changes) [2].

### A.3   Connection to self-supervised learning

The high-level concept of Self-SL pre-training is to obtain a model by training it with a proxy task that is not dependent on annotations, leading to representations that generalize well to a specific task. This allows the induction of information from unlabeled data into the model in the form of an initialization, which can be interpreted as a form of bias. Usually, these representations are supposed to be clustered based on some form of similarity, which is often induced directly by the proxy task and also the reason why different proxy tasks are useful for different downstream tasks. Several different Self-SL pre-training strategies were developed based on different tasks s.a. generative models or clustering [12, 13, 21, 25], with contrastive training being currently the de facto standard in image classification. For a more thorough overview over Self-SL we refer the interested reader to [41]. Based on this, we use the popular contrastive SimCLR [13] training strategy as a basis for our Self-SL pre-training.

### A.4   Connection to semi-supvervised learning

In Semi-SL the core idea is to regularize $p(Y|x,\theta)$ by inducing information about the structure of $p(x)$ using the unlabeled pool additionally to the labeled dataset. Usually, this leads to the representations of unlabeled samples with the clustering being more in line with the structure of the supervised task [39]. Several different Semi-SL methods were developed based on regularizations on unlabeled samples, which often fall into the category of enforcing consistency of predictions against perturbations and/or reducing the uncertainty of predictions (for more information, we refer to [45]). For a more thorough overview of Semi-SL we refer interested readers to [7, 45] In our experiments, we use FixMatch [52] as a Semi-SL method, which combines both aforementioned principles of consistency and uncertainty reduction in a simple manner.

# B Active learning literature, in more detail

We will discuss the current literature landscape of deep active classification with a focus on our proposed key-pitfalls as shown in Figure 1b.

The rules for evaluation of each of the five pitfalls (P1-P5) are:

| | |
|---|---|
| *P1 Data distribution* | Use of multiple datasets for evaluation featuring class-imbalanced datasets. |
| *P2 Starting budget* | Evaluation or ablating the influence of the starting budget on multiple datasets explicitly. |
| *P3 Query size* | Evaluation or ablating the influence of the query size on multiple datasets explicitly. |
| *P4 Performant baselines* | Performance is close to ours or Munjal et al. [44] for ST models on CIFAR-10/100(see Appendix H for details).[3] |
| *P4 HP Optim. & Val. Split* | The use of a dedicated validation set to configure the classifier. |
| *P5 Self-SL* | Benchmarking AL with a performant Self-SL training paradigms. |
| *P5 Semi-SL* | Benchmarking AL with a performant Semi-SL training paradigm |

**Munjal et al. [44]**   Evaluate the performance of AL methods and compare against and with well finetuned baseline models using AutoML.

*P1 Data Distribution:* Perform experiments on CIFAR-10, CIFAR-100 and limited experiments on ImageNet. They perform an ablation on CIFAR-100 with an artificial imbalanced dataset.
→ (✔) due to limited imbalanced datasets.

*P2 Starting Budget:* Perform no experiments at all regarding the starting budget.
→ **X**

*P3 Query Size:* Perform ablations on CIFAR-10/100 comparing query sizes of 5% (2500) to 10% (5000). → ✔

*P4 Performant Baselines:* They achieve performance on CIFAR-10/100 on par with ours (see Appendix H). → ✔

*P4 HP Optim. & Val. Split* They explicitly use a validation set and finetune their hyperparameters based on the validation set performance using AutoML.
→ ✔

*P5 Self-SL:* They do not consider using models pre-trained with Self-SL.
→ **X**

*P5 Semi-SL:* They do not consider using models trained with semi-supervised training paradigms.
→ **X**

**Mittal et al. [43]**   Evaluate the performance of AL methods and set them into context with semi-supervised training paradigms.

*P1 Data Distribution:* Perform experiments on CIFAR-10, CIFAR-100.
→ **X**

*P2 Starting Budget:* Perform experiments both on the standard setting with starting budget of 5000 (10%) on CIFAR-10/100 as well as 250 (CIFAR-10) and 500 (CIFAR-100).
→ (✔) due to limited settings.

*P3 Query Size:* They do not consider speicifally ablating the query size.
→ **X**

---

[3]We only base this on ST models, as getting good performance with Self-SL and Semi-SL for low data settings can be achieved without taking HP configuration into account, as they can simply be taken from a paper focusing on them which often use very large validation sets.

*P4 Performant Baselines:* Their random baseline is more performant on CIFAR10/100 than most of the literature. However, not as good as Munjal et al. [44] or ours (see Appendix H).
→ (✔) due to performance being good but no on par with ours.

*P4 HP Optim. & Val. Split* They do not state optimizing their hyperparameters based on a validation set.
→ X

*P5 Self-SL:* They do not consider using models pre-trained with Self-SL.
→ X

*P5 Semi-SL:* They evaluate AL with and against the semi-supervised training paradigm 'Unsupervised Data Augmentation for Consistency Training'.
→ ✔

**Bengar et al. [6]** Evaluate the performance of AL methods and set them into context with self-supervised training paradigms.

*P1 Data Distribution:* They perform experiments on CIFAR-10/100 and TinyImageNet. All of which are class balanced datasets.
→ X due to only evaluating balanced datasets.

*P2 Starting Budget:* They use 3 different starting budgets on each of their three datasets. CIFAR-10: 0.1%, 1%, 10%; CIFAR-100: 1%, 2%, 10%; Tiny ImageNet: 1%, 2%, 10% (% of the whole dataset).
→ ✔

*P3 Query Size:* Each of the three different starting budgets has a different query size resulting in overlapping experiments. Therefore it would be possible to draw some conclusions about the influence of the query size.
→ (✔) due missing selective evaluation of query size.

*P4 Performant Baselines:* Their supervised random baseline is performing worse on CIFAR-10/100 than most models in the literature (see Appendix H).
→ X

*P4 HP Optim. & Val. Split:* They do not state optimizing their hyperparameters based on a validation set.

*P5 Self-SL:* They evaluate AL methods with and against one self-supervised training paradigm (SimSiam).
→ ✔

*P5 Semi-Supervise Learning:* They do not consider using models trained with semi-supervised training paradigms.
→ X

**Gao et al. [20]** Evaluate the performance of AL methods against and in the context of semi-supervised training paradigms. Further, they propose a new query method designed for AL with models that are trained with a Semi-SL training paradigm.

*P1 Data Distribution:* They perform experiments on CIFAR-10/100 and ImageNet.
→ X due to only evaluating balanced datasets.

*P2 Starting Budget:* They perform a specific ablation about the importance of the starting budget on CIFAR-10 with multiple settings and discuss it.
→ ✔

*P3 Query Size:* In addition to the standard experiments, they perform experiments with query sizes of 50 and 250. However, they do not specifically discuss its importance.
→ (✔) due missing selective evaluation of query size.

*P4 Performant Baselines:* The performance of their supervised random baseline models in the main comparison is not close to our performance on CIFAR-10/100.
→ **X**

*P4 HP Optim. & Val. Split:* They do not state optimizing their hyperparameters based on a validation set.
→ **X**

*P5 Self-SL:* They do not consider using models pre-trained with Self-SL.
→ **X**

*P5 Semi-SL:* They evaluate AL with and against the semi-supervised training paradigm (MixMatch).
→ ✔

**Yi et al. [54]**   They propose to use self-supervised pre-text as a basis for query functions.

*P1 Data Distribution:* Perform experiments on CIFAR-10, an imbalanced version of CIFAR-10, Caltech-101 and ImageNet.
→ ✔

*P2 Starting Budget:* One experiment is performed where they select the starting budget with their proposed Active Learning method on CIFAR-10. Otherwise, they do not evaluate the performance under different starting budgets.
→ **X**

*P3 Query Size:* They do not evaluate the performance with regard to different query sizes.
→ **X**

*P4 Performant Baselines:* Their supervised random baseline models are not close to the performance of our random baseline models on CIFAR-10 (see Appendix H).
→ **X**

*P4 HP Optim. & Val. Split:* They do not state optimizing their hyperparameters based on a validation set.
→ **X**

*P5 Self-SL:* They consider several different Self-SL paradigms ('Rotation Prediction', 'Colorization', 'Solving jigsaw puzzles' and 'SimSiam') based on which they select rotation prediction for their experiments.
→ (✔) due to selection of non state-of-the-art Self-SL paradigm.

*P5 Semi-Supervise Learning:* They do not consider using models trained with semi-supervised training paradigms.
→ **X**

**Krishnan et al. [35]**   They propose to use a supervised contrastive training paradigm as a basis for two AL methods.

*P1 Data Distribution:* Perform experiments on Fashion-MNIST, SVHN and CIFAR-10 and an imbalanced version of CIFAR-10.
→ (✔) due to imbalanced CIFAR-10 being simulated.

*P2 Starting Budget:* They do not evaluate the performance under different starting budgets.
→ **X**

*P3 Query Size:* They do not evaluate the performance with regard to different query sizes.
→ **X**

*P4 Performant Baselines:* Their supervised random baseline models are not close to the performance of our random baseline models on CIFAR-10 (see Appendix H).
→ **X**

*P4 HP Optim. & Val. Split:* They do not state optimizing their hyperparameters based on a validation set.
→ **X**

*P5 Self-SL:* They do not consider using models pre-trained with Self-SL.
→ **X**

*P5 Semi-Supervise Learning:* They do not consider using models trained with semi-supervised training paradigms.
→ **X**

**Kim et al. [30]**   They propose task-aware active learning which is a combination of learning loss active learning and variational adversarial active learning.

*P1 Data Distribution:* Perform experiments on CIFAR-10, CIFAR-100, CALTECH 101 and imbalanced CIFARS.
→ ✔

*P2 Starting Budget:* They do not evaluate the performance under different starting budgets.
→ (✔)

*P3 Query Size:* They do not evaluate the performance with regard to different query sizes.
→ (✔)

*P4 Performant Baselines:* Their supervised random baseline models perform good but not on par with ours on CIFAR-10 and 100.
→ (✔)

*P4 HP Optim. & Val. Split:* They do not state optimizing their hyperparameters based on a validation set.
→ **X**

*P5 Self-SL:* They do not consider using models pre-trained with Self-SL.
→ **X**

*P5 Semi-SL:* They do not consider using models trained with semi-supervised training paradigms.
→ **X**

**Beck et al. [4]**   Evaluate several AL methods in different settings to gain an understanding which AL methods outperform random queries. Further, they provide the Al toolkit DISTIL.

*P1 Data Distribution:* Perform experiments on CIFAR-10, CIFAR-100, Fashion-MNIST, SVHN and MNIST. → **X** due to no class imbalance.

*P2 Starting Budget:* They perform one experiment, where they evaluate a lower starting budget for MNIST.
→ (✔) due limited dataset.

*P3 Query Size:* They evaluate three different query sizes on CIFAR-10, but do so only for Random, Entropy and BADGE.
→ (✔) due to limited scope.

*P4 Performant Baselines:* Their supervised random baseline models perform good but no par with our random baseline models on CIFAR-10/100 (see Appendix H).
→ (✔) due to limited scope.

*P4 HP Optim. & Val. Split:* They do not state optimizing their hyperparameters based on a validation set.
→ **X**

*P5 Self-SL:* They do not consider using models pre-trained with Self-SL.
→ **X**

*P5 Semi-SL:* They do not consider using models trained with semi-supervised training paradigms.
→ **X**

**Zhan et al. [58]**   Evaluate a multitude of different AL methods and provide the AL toolkit *DeepAL⁺*.

*P1 Data Distribution:* Perform experiments on Tiny ImageNet, CIFAR-10 (and CIFAR-10 imbalanced), CIFAR-100, Fashion-MNIST, EMNIST and SVHN. Further Experiments are performed on an Histopathological image Classification Task (BreakHis) and Chest X-Ray Pneumonia classification (Pneumonia-MNIST) as well as the Waterbird dataset adopted from object recognition with correlated backgrounds.
→ 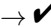

*P2 Starting Budget:* They do not evaluate the performance under different starting budgets.
→ **X**

*P3 Query Size:* They evaluate multiple different query sizes on CIFAR-10 and analyze the difference.
→ 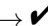

*P4 Performant Baselines:* Their supervised random baseline models are not close to the performance of our random baseline models on CIFAR-10/100.
→ **X**

*P4 HP Optim. & Val. Split:* They do not state optimizing their hyperparameters based on a validation set.
→ **X**

*P5 Self-SL:* They do not consider using models pre-trained with Self-SL.
→ **X**

*P5 Semi-SL:* They do not consider using models trained with semi-supervised training paradigms.
→ **X**

**Chan et al. [11]**   Evaluate how AL methods interact with self- and semi-supervised training paradigms and how and whether they yield a benefit. The experiments in this paper differ from standard AL experiments by using only one query cycle, making it hard to compare systematically.

*P1 Data Distribution:* Perform experiments on CIFAR-10 and CIFAR-100.
→ **X**

*P2 Starting Budget:* Experiments are performed with a fixed starting budget of 3 samples per class or 2 samples per class in one case. This does not allow for evaluate of the influence of the starting budget on AL methods.
→ **X**

*P3 Query Size:* They query all samples for their final performance in one query step. This does not allow for evaluation of the influence of the query size on AL methods.
→ **X**

*P4 Performant Baselines:* Their supervised random baseline models perform on par with our random baseline models on CIFAR-10/100.
→ ✔

*P4 HP Optim. & Val. Split:* They do not state optimizing their hyperparameters based on a validation set.
→ **X**

*P5 Self-SL:* They use Debiased Contrastive Learning as self-supervised pretext task.
→ ✔

*P5 Semi-SL:* They use Pseudo-labeling and FixMatch as semi-supervised training paradigms.
→ ✔

Table 1: Number of Samples for each class in CIFAR-10 LT dataset. Validation and test sets are balanced.

| Class | Train Split |
|---|---|
| airplane | 4500 |
| automobile (but not truck or pickup truck) | 2913 |
| bird | 1886 |
| cat | 1221 |
| deer | 790 |
| dog | 512 |
| frog | 331 |
| horse | 214 |
| ship | 139 |
| truck (but no pickup truck) | 90 |

## C   Dataset details

Each dataset is split into a training, a validation and a test split.

For CIFAR-10/100 (LT) datasets the test split of size 10000 observations is already given and for MIO-TCD and ISIC-2019 we use a custom test split of 25% random observations of the entire dataset size. For MIO-TCD and ISIC-2019 the train, validation and test splits are imbalanced.

The validation split for all CIFAR-10 and CIFAR-100 datasets are 5000 randomly drawn observations corresponding to 10% of the entire dataset. For CIFAR-10 LT the validation split also consists of 5000 samples obtained from the dataset before the long-tail distribution is applied onto the training split. The CIFAR-10 LT validation split is therefore balanced. For MIO-TCD and ISIC-2019 the validation splits consist of 15% of the entire dataset.

The shared training & pool dataset for CIFAR-10/100 consists of 45000 observations. For CIFAR-10 LT the training & pool datasets consist of 12,600 observations. For MIO-TCD and ISIC-2019 the training & pool datasets consist of 60% the dataset.

### C.1   Dataset descriptions

1. CIFAR-10: natural images containing 10 classes, label distribution is uniform
   Splits: (Train:45000; Val: 5000; Test; 10000)
   Whole Dataset: 60000

2. CIFAR-100: natural images containing 100 classes, label distribution is uniform
   Splits: (Train:45000; Val: 5000; Test; 10000)
   Whole Dataset: 60000

3. CIFAR-10 LT: natural images containing 10 classes, label distribution of test and validation split is uniform, label distribution of train split is artifically altered with imbalance factor $\rho = 50$ according to [10]. The resulting label distribution is shown in Tab. 1.
   Splits: (Train:$\sim$12,600; Val: 5000; Test; 10000)
   Whole Dataset: 27600

4. ISIC-2019: dermoscopic images containing 8 classes, label distribution of the dataset is imbalanced and shown in Tab. 2
   Splits: (Train:15200; Val: 3799; Test; 6332)
   Whole Dataset: 25331

5. MIO-TCD: natural images of traffic participants containing 11 classes, label distribution of the dataset is imbalanced and shown in Tab. 3
   Splits: (Train:311498; Val: 77875; Test; 129791)
   Whole Dataset: 519164

Table 2: Number of Samples for each class in ISIC-2019

| Class | Whole Dataset |
|---|---|
| Melanoma | 4522 |
| Melanocytic nevus | 12875 |
| Basal cell carcinoma | 3323 |
| Benign keratosis | 867 |
| Dermatofibroma | 197 |
| Vascular lesion | 63 |
| Squamos cell carcinoma | 64 |

Table 3: Number of samples for each class in MIO-TCD

| Class | Whole Dataset |
|---|---|
| Articulated Truck | 10346 |
| Background | 16000 |
| Bicycle | 2284 |
| Bus | 10316 |
| Car | 260518 |
| Motorcycle | 1982 |
| Non-motorized vehicle | 1751 |
| Pedestrian | 6262 |
| Pickup truck | 50906 |
| Single unit truck | 5120 |
| Work van | 9679 |

# D  Experimental setup, in more detail

Here we detail the most crucial information for reprocubility, re-implementation and checking our implementation. When in doubt, trust the information documented here with regard to what we wanted to do in our code.

## D.1  Initial dataset setup

Before we do anything else the datasets are split according to Figure 4 resulting in a tain split, a validation split and a test split. Each dataset has 3 different validation splits while always using the same test split. This is to ensure comparability across these splits without relying on cross-validation. The exact splits for each dataset are detailed in Appendix C. After that the final datasets use for training and validation are then labeled according to the 'label strategy', which is described in Figure 5. For all balanced datasets, we use class balanced label strategies since the label strategy only leads to different outcomes for imbalanced datasets. For CIFAR-10 LT we use the label strategy on the train split only, whereas for MIO-TCD and ISIC-2019 we use the label strategy on both train and validation split. The amount of data which is labeled for the final datasets of each split is then dependent upon the label-regime (described in more detail in Appendix D.2).

## D.2  Label regimes

The exact label regimes are obtained by first taking the corresponding splits and then using the proper label strategy (see Figure 5) in combination with the starting budget and validation set size according to Tab. 4.

## D.3  Model architecture and training

On each training step the model is trained from its initialization to avoid a 'mode collapse' [32]. Further we select the checkpoint with the best validation set performance in the spirit of [19]. A ResNet-18 [24] is the backbone for all of our experiments with weight decay disabled on bias parameters. If not otherwise noted, a nesterov momentum optimizer with momentum of 0.9 is used.

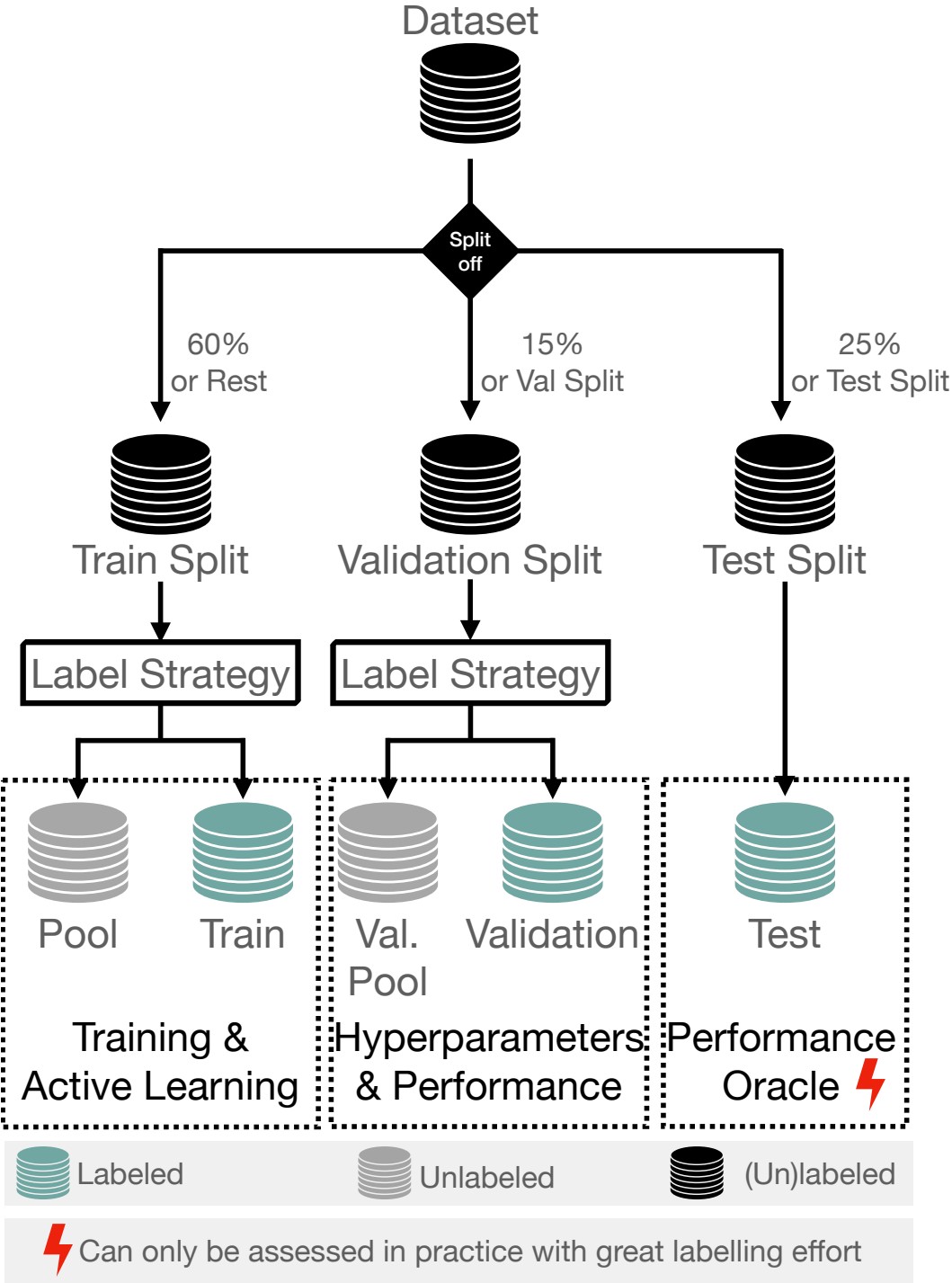

Figure 4: Description of the three different data splits and their use-cases. The complete separation of a validation split allows to compare across label regimes and incorporate techniques for performance evaluation s.a. Active Testing [34]. For evaluation and development the test split should be as big as possible since QM recommendations are based on the test set performance making it a form of "oracle". An estimate of the size a dataset is required to have to measure specific performance differences can be derived using Hoeffding's inequality [26, 45].

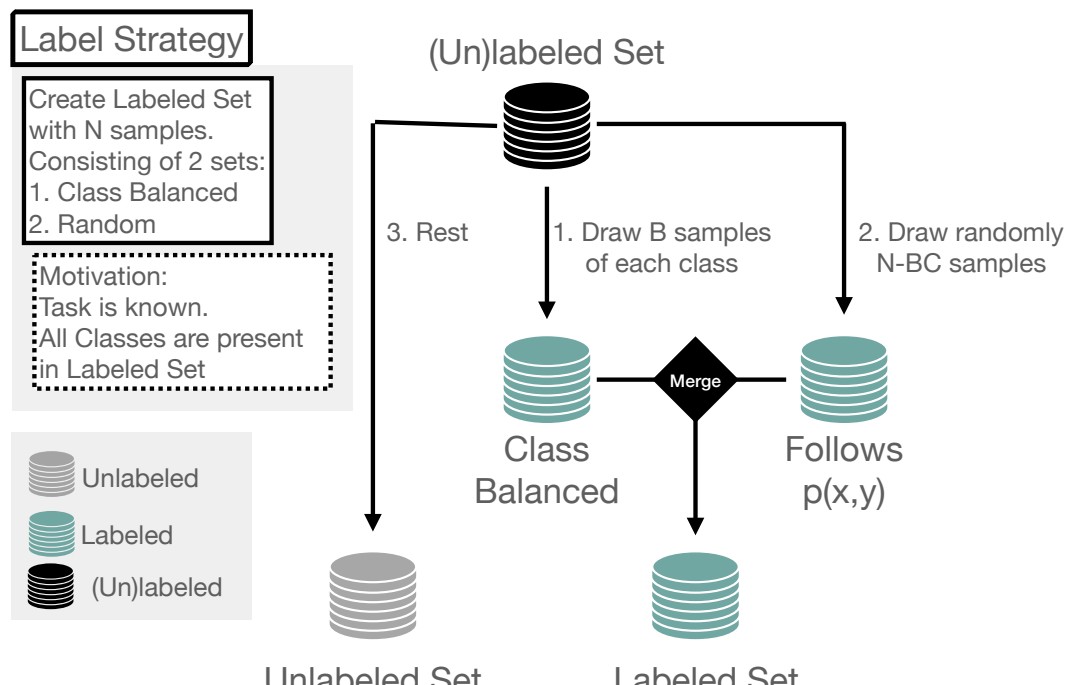

Figure 5: The Label Strategy used on the two roll-out datasets MIO-TCD and ISIC-2019 and for train and pool set on CIFAR-10 LT. For class balanced datasets this strategy does not induce meaningful changes to balanced starting budgets.

Table 4: The exact values for all label regimes. Final Budget denotes the amount of labeled training samples at the end of the AL pipeline.

| Dataset | CIFAR-10 | | | CIFAR-100 | | | CIFAR-10 LT | | | MIO-TCD | | | ISIC-2019 | | |
| Label Regime | Low | Mid | High | Low | Mid | High | Low | Mid | High | Low | Mid | High | Low | Mid | High |
|---|---|---|---|---|---|---|---|---|---|---|---|---|---|---|---|
| Starting Budget | 50 | 250 | 1000 | 500 | 1000 | 5000 | 50 | 250 | 1000 | 55 | 275 | 1100 | 40 | 200 | 800 |
| Query Size | 50 | 250 | 1000 | 500 | 1000 | 5000 | 50 | 250 | 1000 | 55 | 275 | 1100 | 40 | 200 | 800 |
| Final Budget | 500 | 2500 | 10000 | 5000 | 10000 | 25000 | 500 | 2500 | 10000 | 550 | 2750 | 11000 | 400 | 2000 | 8000 |
| Validation Set Size | 250 | 1250 | 5000 | 2500 | 5000 | 5000 | 250 | 1250 | 5000 | 275 | 1375 | 5500 | 200 | 800 | 3799 |

For Self-SL models we use a two layer MLP as a classification head to make better use of the Self-SL representations with further details in Appendix D.5. To obtain bayesian models we add dropout on the final representations before the classification head with probability ($p = 0.5$) following [19]. For all experiments on imbalanced datasets, we use the weighted CE-Loss following [44] based on the implementation in SK-Learn [9] if not otherwise noted. Models trained purely on the labeled dataset upsample it to a size of 5500 following [32] if the labeled train set is smaller.

**Bayesian Models** All steps that require bayesian properties of the models including the prediction are obtained by drawing 50 MC samples following [19, 32].

**ST** ST models are trained for 200 epochs with Cosine Annealing and 10 epochs warmup.

**Self-SL** Self-SL pre-trained models are trained for 80 epochs with a reduction of the learning rate with a factor of 10 every 20 epochs (MultiStepLR) and using a Mulit-Layer-Percpeptron (MLP) classification head (detailed description in Appendix D.5). The complete setup of the training for SimCLR is described in Appendix D.4.

**Semi-SL** Semi-SL training is identical to the one proposed with the FixMatch method [52], except that we do not use exponentially moving average models and restrict the training step from 1e6 to 2e5. The FixMatch implementation in our experiments is based on the open-source implementation

Table 5: HPs of the SimCLR pre-text training on each dataset. HP for CIFAR datasets are directly taken [13] whereas MIO-TCD and ISIC-2019 HP are adapted from ImageNet experiments.

| Dataset | CIFAR-10/CIFAR100/CIFAR-10 LT | MIO-TCD | ISIC-2019 |
|---|---|---|---|
| Epochs | 1000 | 200 | 1000 |
| Optimizer | LARS | LARS | |
| Scheduler | Cosine Annealing | Cosine Annealing | |
| Warmup Epochs | 10 | 10 | |
| Temperature | 0.5 | 0.1 | |
| Batch Size | 512 | 256 | |
| Learning Rate | 1 | 0.3 | |
| Weight Decay | 1E-4 | 1E-6 | |
| Transform. Gauss Blur | False | True | |
| Transform. Color Jitter | Strength=0.5 | Strength=1.0 | |

Table 6: MLP Head Ablation for Self-SL models on CIFAR-10, over all labeled training set a small improvement for Multi-Layer-Perceptron is measurable compared to Linear classification head models. Reported as mean (std).

| Labeled Train Set | Classification Head | Accuracy (Val) % | Accuracy (Test) % |
|---|---|---|---|
| 50 | Linear | 69.87(1.62) | 69.90(2.18) |
| 50 | 2 Layer MLP | 71.47(3.06) | 71.54(0.56) |
| 500 | Linear | 84.67(0.36) | 83.51(0.45) |
| 500 | 2 Layer MLP | 85.37(0.16) | 84.60(0.37) |
| 1000 | Linear | 87.13(0.69) | 85.97(0.64) |
| 1000 | 2 Layer MLP | 87.69(0.55) | 86.57(0.42) |
| 5000 | Linear | 90.77(0.44) | 90.20(0.21) |
| 5000 | 2 Layer MLP | 91.12(0.32) | 90.25(0.24) |

of [4] and MixMatch for distribution alignment [5]. We always select the final Semi-SL model of the training for testing and querying. On imbalanced datasets we change the supervised term to the weighted CE-Loss and use distribution alignment on every dataset except for CIFAR-10 (where it does not improve performance [52]). The HP sweep for our Semi-SL models includes weight decay and learning rate.

**Hyperparameters** All information with regard to the final HPs and our proposed methodology of finding them is detailed in Appendix E

### D.4 Self-supervised SimCLR pre-text training

Our implementation wraps the Pytorch-Lightning-Bolts implementation of SimCLR: https://lightning-bolts.readthedocs.io/en/latest/models/self_supervised.html#simclr . The training of our SimCLR models is performed by excluding the validation splits. Therefore three models are trained on each dataset, one for each different validation split. In Tab. 5 we give a list of the HPs used on each of our five different datasets. All other HPs are taken from [13]. Further, we did not optimize the HPs for SimCLR at all, meaning that on MIO-TCD and ISIC-2019 Self-SL models could perform even better than reported here.

### D.5 MLP head for self-supervised pretrained models

The MLP Head used for the Self-SL models has 1 hidden layer of size 512 uses ReLU nonlinearities and BatchNorm. The results on CIFAR-10 based on which this design decision is based on is shown in Tab. 6.

---

[4]https://github.com/kekmodel/FixMatch-pytorch
[5]https://github.com/google-research/mixmatch

### D.6 List of data transformations

**Standard** The standard augmentations we use are based on the different datasets.
For CIFAR datasets these are in order of execution: RandomHorizontalFlip, RandomCrop to 32x32 with padding of size 4.

For MIO-TCD we use the standard ImageNet transformations: RandomResizedCrop to 224x224, Random Horizontal Flip.

For ISIC-2019 we use ISIC transformations which are: Resize to 300x300, RandomHorizontalFlip, RandomVerticalFlip, ColorJitter(0.02, 0.02, 0.02, 0.01) ,RandomRotation(rotation=(-180, 180), translate=(0.1, 0.1), scale=(0.7, 1.3)), RandomAffine(-180, 180), RandomCrop to 224x224.
These are based on the ISIC-2018 challenge best single model submission:
https://github.com/JiaxinZhuang/Skin-Lesion-Recognition.Pytorch

**RandAugmentMC** We use the same set of image transformations used in RandAugment [17] with the parameters N=2 and M=10. A detailed list of image transformations alongside the corresponding values can be seen in [52] (Table 12).

The RandAugmentMC transformations were used additionally after the corresponding standard transformations for each dataset. RandAugmentMC(CIFAR) also adds cutout as a final transformation.

**RandAugmentMC weak** Works identical as RandAugmentMC and uses the same set of image transformations as for RandAugmentMC but changed its parameters to N=1 and M=2. Therefore the maximal range of values is divided by a factor of 5.

RandAugmentMC weak does not use cutout in difference to RandAugmentMC on CIFAR datasets.

### D.7 Performance measure

As a measure of performance on CIFAR-10, CIFAR-100 and CIFAR-10 LT we use the accuracy while on MIO-TCD and ISIC-2019 we use balanced accuracy which is identical to mean recall shown in Equation (3).

$$\text{Mean Recall} = \sum_{c=1}^{C} \frac{1}{C} \frac{\text{TP}_c}{\text{TP}_c + \text{FN}_c} \tag{3}$$

Where $C$ denotes the number of classes $\text{TP}_c$ is the number of true positives for class $c$ and $\text{FN}_c$ being the number of samples belonging to class $c$ being wrongly misclassified as another class.

### D.8 Computational effort

Experiments were executed on a Cluster with access to multiple NVIDIA graphics cards. All ST and Self-SL pre-trained experiments used a single Nvidia RTX 2080 (10.7GB video-ram) graphic cards except for the BADGE experiments on CIFAR-100 and MIO-TCD which required more video-ram using Nvidia Titan RTX (23.6GB video-ram). The Semi-SL models on the CIFAR-10/100 (LT) datasets used also a single Nvidia RTX 2080 while on MIO-TCD and ISIC-2019 the Nvidia Titan RTX was utilized. For the results in our main table (excluding the HP optimization), the overall runtime was:

- All ST experiments: 1800 GPU hours
- All Self-SL pre-trained experiments: 1350 GPU hours[6]
- All Semi-SL experiments[7]: 11200 GPU hours

## E   Proposed hyperparameter optimization

Our proposed HP optimization for AL is based on the notion of minimizing HP selection effort by simplifying and reducing the search space. We use SGD Optimizer with Nesterov momentum of 0.9

---

[6]Excluding the pre-training
[7]For only 2 label regimes and exluding MIO-TCD and ISIC-2019

Table 7: Final HPs for each dataset and label regime for our ST models based on our HP tuning. HPs denoted with a * are fixed across datasets and HP denoted with a + are pre-selected for each dataset while all other HP are obtained via sweeping.

| Dataset | CIFAR-10 | | | CIFAR-100 | | | CIFAR-10 LT | | | MIO-TCD | | | ISIC-2019 | | |
|---|---|---|---|---|---|---|---|---|---|---|---|---|---|---|---|
| Label Regime | Low | Mid | High | Low | Mid | High | Low | Mid | High | Low | Mid | High | Low | Mid | High |
| Epochs* | | 200 | | | 200 | | | 200 | | | 200 | | | 200 | |
| Optimizer* | | SGD Nesterov 0.9 | | | SGD Nesterov 0.9 | | | SGD Nesterov 0.9 | | | SGD Nesterov 0.9 | | | SGD Nesterov 0.9 | |
| Scheduler* | | Cosine Annealing | | | Cosine Annealing | | | Cosine Annealing | | | Cosine Annealing | | | Cosine Annealing | |
| Warmup Epochs* | | 10 | | | 10 | | | 10 | | | 10 | | | 10 | |
| Loss* | | CE-Loss | | | CE-Loss | | | CE-Loss | | | CE-Loss | | | CE-Loss | |
| Sampling+ | | standard | | | standard | | | oversampling | | oversampling | | | oversampling | | |
| Batch Size+ | | 1024 | | | 1024 | | | 1024 | | | 512 | | | 512 | |
| Learning Rate | | 0.1 | | | 0.1 | | | 0.1 | | 0.01 | 0.01 | 0.1 | | | |
| Weight Decay | | 5E-3 | | | 5E-3 | | | 5E-3 | | 5E-3 | 5E-4 | 5E-3 | | 5E-3 | |
| Data Augmentation | | RandAugmentMC (CIFAR) | | | RandAugmentMC (CIFAR) | | | RandAugmentMC (CIFAR) | | | RandAugmentMC (ImageNet) | | | RandAugmentMC (ISIC) | |

Table 8: Final HPs for each dataset and label regime for our Self-SL models based on our HP tuning. Overall Performance was remarkably stable with regard to HPs and stronger augmentations did not necessarily improve performance in the same way as for ST models. This is presumably due to the pre-trained representations. HP denoted with a * are fixed across datasets and HP denoted with a + are pre-selected for each dataset while all other HP are obtained via sweeping.

| Dataset | CIFAR-10 | | | CIFAR-100 | | | CIFAR-10 LT | | | MIO-TCD | | | ISIC-2019 | | |
|---|---|---|---|---|---|---|---|---|---|---|---|---|---|---|---|
| Label Regime | Low | Mid | High | Low | Mid | High | Low | Mid | High | Low | Mid | High | Low | Mid | High |
| Epochs* | | 80 | | | 80 | | | 80 | | | 80 | | | 80 | |
| Optimizer* | | SGD Nesterov 0.9 | | | SGD Nesterov 0.9 | | | SGD Nesterov 0.9 | | | SGD Nesterov 0.9 | | | SGD Nesterov 0.9 | |
| Scheduler* | | MulitStepLR | | | MulitStepLR | | | MulitStepLR | | | MulitStepLR | | | MulitStepLR | |
| Warmup Epochs* | | 0 | | | 0 | | | 0 | | | 0 | | | 0 | |
| Loss* | | CE-Loss | | | CE-Loss | | | CE-Loss | | | CE-Loss | | | CE-Loss | |
| Sampling+ | | standard | | | standard | | | oversampling | | oversampling | | | oversampling | | |
| Batch Size+ | | 64 | | | 64 | | | 64 | | | 256 | | | 128 | |
| Learning Rate | | 0.001 | | | 0.001 | | 0.01 | 0.01 | 0.001 | | 0.001 | | 0.001 | 0.001 | 0.01 |
| Weight Decay | | 5E-3 | | | 5E-3 | | 5E-4 | 5E-4 | 5E-3 | | 5E-3 | | 5e-3 | 5E-4 | 5e-3 |
| Data Augmentation | | RandAugmentMC weak (CIFAR) | | | RandAugmentMC weak (CIFAR) | | | Standard (CIFAR) | | RandAugmentMC weak (ImageNet) | Standard (ImageNet) | Standard (ImageNet) | | Standard (ISIC) | |

and select a number of epochs that always allow a complete fit of the model. The scheduler is also fixed across experiments as are the warmup epochs if used. Secondly, we pre-select the batchsize for each dataset since it is usually not a critical HP as long as it is big enough for BatchNorm to work properly.

**ST** For our ST models the final HP for each dataset and label regime are shown in Tab. 7.
HP sweep: weight decay: (5E-3, 5E-4); learning rate: (0.1, 0.01); data transformation: (RandAugmentMC, Standard)

**Self-SL** For our Self-SL pre-trained models the final HP for each dataset and label regime are shown in Tab. 8.
HP sweep: weight decay: (5E-3, 5E-4); learning rate: (0.01, 0.001); data transformation: (RandAugmentMC weak, Standard)

**Semi-SL** For our Semi-SL models we follow [52] with regard to HP selection as closely as possible. The final HP for each dataset and label regime are shown in Tab. 9.
HP sweep: weight decay and learning rate.

Table 9: Final HPs for each dataset and label regime for our Semi-SL models based on our HP tuning. HP denoted with a * are fixed across datasets and HP denoted with a + are pre-selected for each dataset while all other HP are obtained via sweeping. – denotes not performed experiments.

| Dataset | CIFAR-10 | | | CIFAR-100 | | | CIFAR-10 LT | | | MIO-TCD | | | ISIC-2019 | | |
|---|---|---|---|---|---|---|---|---|---|---|---|---|---|---|---|
| Label Regime | Low | Mid | High | Low | Mid | High | Low | Mid | High | Low | Mid | High | Low | Mid | High |
| Optimization Steps* | | 2E5 | – | | 2E5 | – | | 2E5 | – | | 2E5 | – | | 2E5 | – |
| Optimizer* | | SGD Nesterov 0.9 | – | | SGD Nesterov 0.9 | – | | SGD Nesterov 0.9 | – | | SGD Nesterov 0.9 | – | | SGD Nesterov 0.9 | – |
| Scheduler* | | Cosine Annealing | – | | Cosine Annealing | – | | Cosine Annealing | – | | Cosine Annealing | – | | Cosine Annealing | – |
| Warmup Steps+ | | 0 | – | | 0 | – | | 0 | – | | 3000 | – | | 3000 | – |
| Loss+ | | CE-Loss | – | | CE-Loss | – | | weigthed CE-Loss | – | | weigthed CE-Loss | – | | weigthed CE-Loss | – |
| Sampling* | | standard | – | | standard | – | | standard | – | | standard | – | | standard | – |
| $\lambda_u^*$ | | 1 | – | | 1 | – | | 1 | – | | 1 | – | | 1 | – |
| $\mu^*$ | | 7 | – | | 7 | – | | 7 | – | | 7 | – | | 7 | – |
| $\tau^*$ | | 0.95 | – | | 0.95 | – | | 0.95 | – | | 0.95 | – | | 0.95 | – |
| Distribution Alignment+ | | False | – | | True | – | | True | – | | True | – | | True | – |
| Batch Size* | | 64 | – | | 64 | – | | 64 | – | | 64 | – | | 64 | – |
| Learning Rate | | 0.03 | – | | 0.03 | – | 0.03 | | – | | | – | | | – |
| Weight Decay | | 5E-4 | – | | 5E-4 | – | 1E-3 | 5E-4 | – | | | – | | | – |
| Data Augmentation+ | | Standard (CIFAR) | – | | Standard (CIFAR) | – | | Standard (CIFAR) | – | | Standard (ImageNet) | – | | Standard (ISIC) | – |
| Unlabeled Augmentation+ | | RandAugmentMC (CIFAR) | – | | RandAugmentMC (CIFAR) | – | | RandAugmentMC (CIFAR) | – | | RandAugmentMC (ImageNet) | – | | RandAugmentMC (ISIC) | – |

# F    Detailed results

## F.1    Main results

**General observations:**    For all datasets, the overall performance of models was primarily determined by the training strategy and the HP selection, with the benefits of AL being generally smaller compared to the proper selection of both. For the three toy datasets, Semi-SL generally performed best, followed by Self-SL and ST last, whereas, for the two real-world datasets, Semi-SL showed no substantial improvement over ST in the first training stage and, therefore, further runs were omitted. Also, the absolute performance gains for Self-SL models with AL are generally smaller compared to ST models. For Semi-SL, there were generally only very small performance gains or substantially worse performance with AL observed. Concerning the effect of AL, the high-label regime proved to work for ST models on all datasets and Self-SL models. On the two real-world datasets, MIO-TCD and ISIC-2019, a dip in performance at  7k samples for all ST models could be observed. This behavior is ablated in Appendix I.1.

**Evaluation using the pair-wise penalty matrix:**    We use the pair-wise penalty matrix (PPM) to compare whether the performance of one query method significantly outperforms the others. It is essentially a measure of how often one method significantly outperforms another method based on a t-test with $\alpha = 0.05$ (more info in [2, 4]). This allows to aggregate results over different datasets and label regimes, with the disadvantage being that the absolute performance is not taken into consideration. When reading a PPM, each row $i$ indicates the number of settings in which method $i$ beats other methods, while column $j$ indicates the number of settings in which method $j$ is beaten by another method.

We show the PPMs aggregated over all datasets and label regimes for each training paradigm in Figure 6.

For all methods, BADGE is the QM that is least often outperformed by other QMs. Further, for Self-SL models, it is never significantly outperformed by Random, whereas it is seldomly significantly outperformed for ST models. Based on this, we deem BADGE to be the best of our compared QMs for both ST and Self-SL models. Since BADGE is more often outperformed by Random (0.5) on the Semi-SL datasets and the additional high training cost for each iteration, we believe that Random is the better choice in many cases.

**Evaluation using the area under the budget curve:**    For each of the following subsections, we added the area under the budget curve (AUBC) for each dataset and label regime to allow assessing the absolute performance each QM brings. Generally, higher values are better. For more information, we refer to [57, 58].

The results on the dataset for AUBC also show that BADGE is always one of the best performing AL methods. This is in line with the findings based on the PPM.

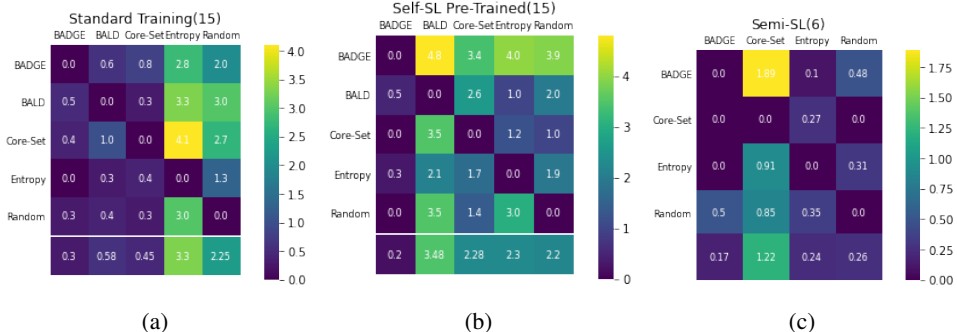

(a)          (b)          (c)

Figure 6: PPMs aggregated over all experiments for Standard Models (a), Self-Sl Pre-Trained Models (b) and Semi-SL models(c).

The value in the title (X) gives the highest possible value in a cell and the lowest row is the mean value across a column $j$ without row $i = j$ signaling how often on average on QM is outperformed by another.

### F.1.1   CIFAR-10

The AUBC values are shown in Tab. 10.

Table 10: Area Under Budget Curve values for CIFAR-10.

| | Label Regime | Low-Label | | Medium-Label | | High-Label | |
| --- | --- | --- | --- | --- | --- | --- | --- |
| | | Mean | STD | Mean | STD | Mean | STD |
| Training | Query Method | | | | | | |
| ST | BADGE | 0.4730 | 0.0105 | 0.7289 | 0.0025 | 0.8599 | 0.0016 |
| | BALD | 0.4744 | 0.0106 | 0.7253 | 0.0051 | 0.8578 | 0.0017 |
| | Entropy | 0.4307 | 0.0018 | 0.6859 | 0.0042 | 0.8498 | 0.0025 |
| | Core-Set | 0.4681 | 0.0038 | 0.7282 | 0.0043 | 0.8629 | 0.0017 |
| | Random | 0.4720 | 0.0144 | 0.7309 | 0.0068 | 0.8526 | 0.0030 |
| Self-SL | BADGE | 0.8282 | 0.0016 | 0.8728 | 0.0018 | 0.9086 | 0.0012 |
| | BALD | 0.8005 | 0.0056 | 0.8692 | 0.0011 | 0.9093 | 0.0010 |
| | Entropy | 0.8002 | 0.0090 | 0.8663 | 0.0017 | 0.9071 | 0.0010 |
| | Core-Set | 0.8224 | 0.0026 | 0.8670 | 0.0009 | 0.9015 | 0.0009 |
| | Random | 0.8117 | 0.0040 | 0.8669 | 0.0009 | 0.8989 | 0.0007 |
| Semi-SL | BADGE | 0.9349 | 0.0010 | 0.9488 | 0.0022 | – | – |
| | Entropy | 0.9193 | 0.0082 | 0.9497 | 0.0007 | – | – |
| | Core-Set | 0.9343 | 0.0018 | 0.9442 | 0.0007 | – | – |
| | Random | 0.9326 | 0.0050 | 0.9478 | 0.0002 | – | – |

**ST**    Results are shown in Figure 7.

**Self-SL**    Results are shown in Figure 8.

**Semi-SL**    Results are shown in Figure 9.

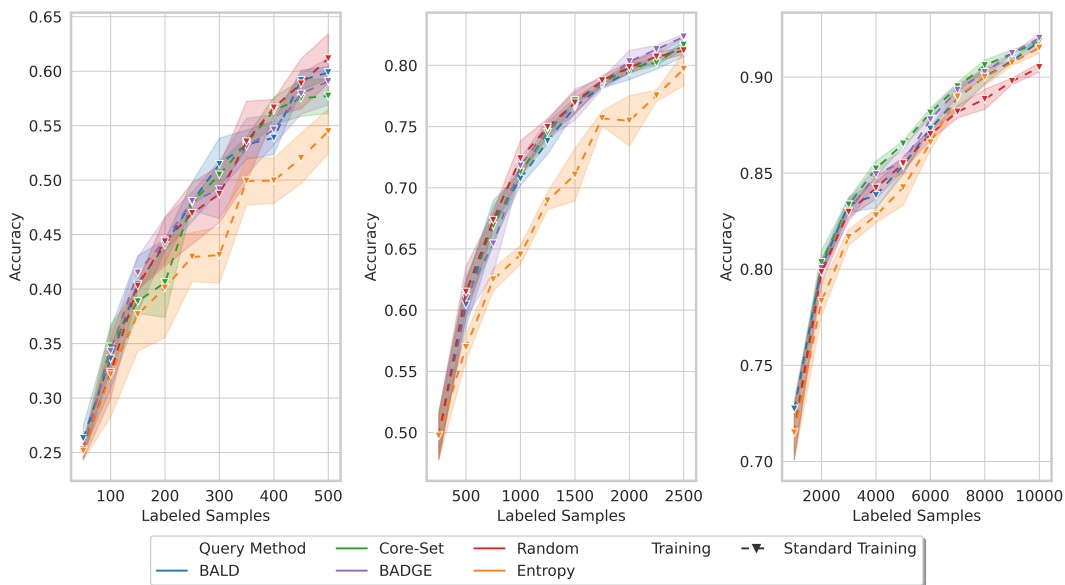

Figure 7: CIFAR-10 ST

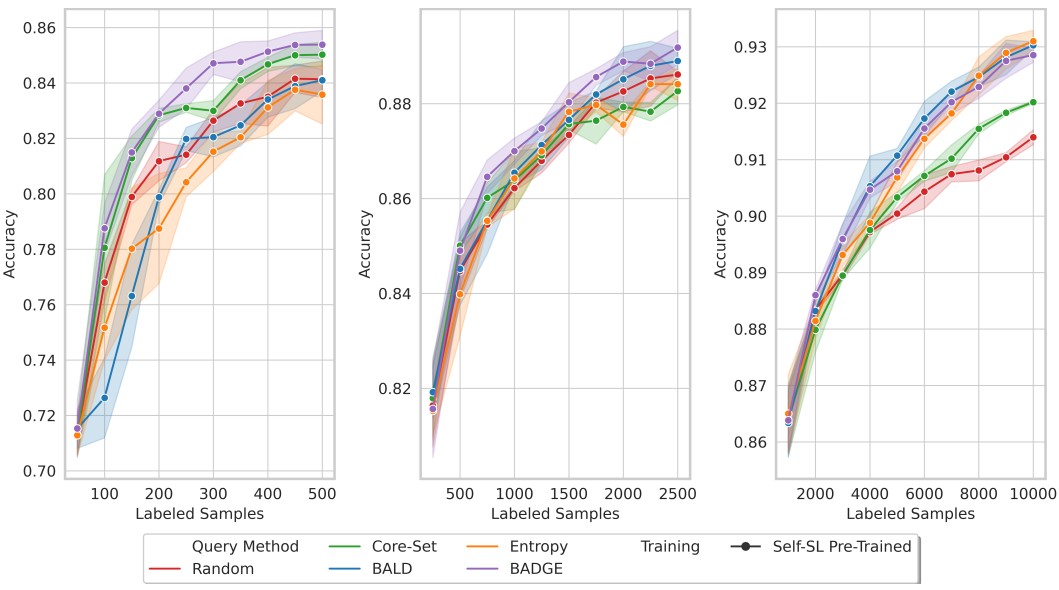

Figure 8: CIFAR-10 Self-SL

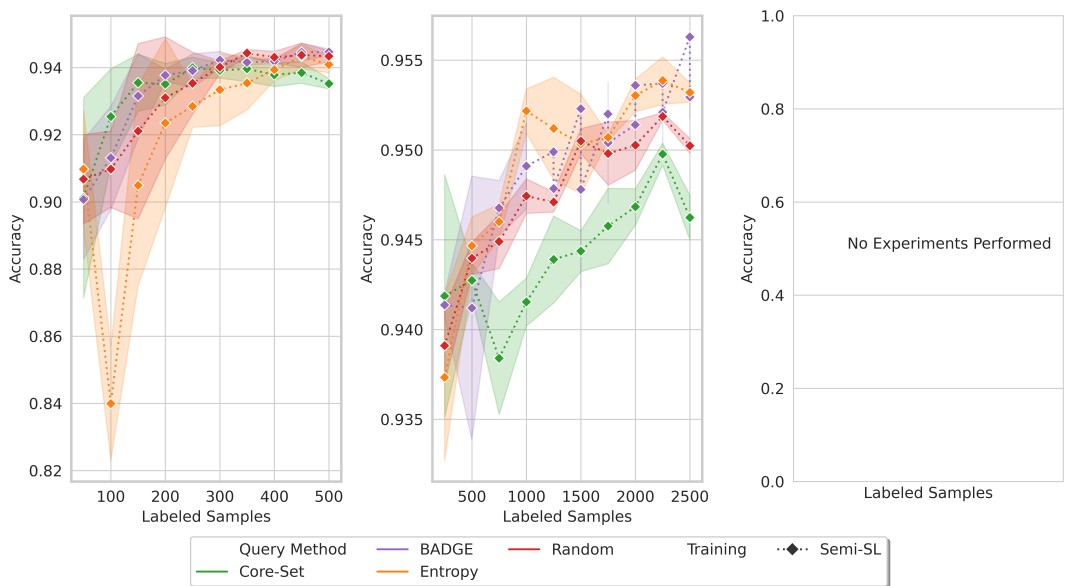

Figure 9: CIFAR-10 Semi-SL

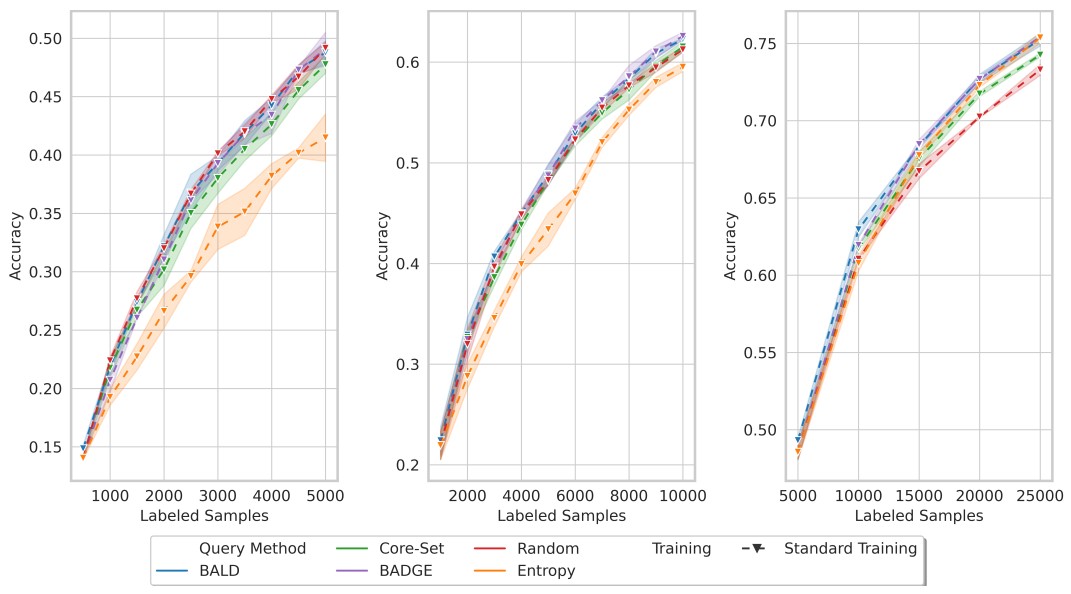

Figure 10: CIFAR-100 ST

## F.1.2 CIFAR-100

The AUBC values are shown in Tab. 11.

Table 11: Area Under Budget Curve values for CIFAR-100.

| Training | Query Method | Low-Label Mean | Low-Label STD | Medium-Label Mean | Medium-Label STD | High-Label Mean | High-Label STD |
|---|---|---|---|---|---|---|---|
| ST | BADGE | 0.3525 | 0.0042 | 0.4855 | 0.0044 | 0.6627 | 0.0017 |
| | BALD | 0.3586 | 0.0007 | 0.4865 | 0.0019 | 0.6658 | 0.0002 |
| | Entropy | 0.3036 | 0.0095 | 0.4440 | 0.0007 | 0.6569 | 0.0021 |
| | Core-Set | 0.3458 | 0.0010 | 0.4767 | 0.0031 | 0.6560 | 0.0004 |
| | Random | 0.3599 | 0.0027 | 0.4791 | 0.0044 | 0.6474 | 0.0015 |
| Self-SL | BADGE | 0.5397 | 0.0030 | 0.6020 | 0.0019 | 0.6858 | 0.0021 |
| | BALD | 0.5028 | 0.0043 | 0.5754 | 0.0027 | 0.6784 | 0.0009 |
| | Entropy | 0.5111 | 0.0066 | 0.5857 | 0.0028 | 0.6857 | 0.0032 |
| | Core-Set | 0.5337 | 0.0044 | 0.5917 | 0.0032 | 0.6804 | 0.0013 |
| | Random | 0.5365 | 0.0017 | 0.5970 | 0.0017 | 0.6757 | 0.0013 |
| Semi-SL | BADGE | 0.5562 | 0.0033 | 0.6222 | 0.0041 | – | – |
| | Entropy | 0.5328 | 0.0158 | 0.6152 | 0.0038 | – | – |
| | Core-Set | 0.5220 | 0.0101 | 0.6083 | 0.0061 | – | – |
| | Random | 0.5713 | 0.0066 | 0.6307 | 0.0016 | – | – |

**ST**  Results are shown in Figure 10.

**Self-SL**  Results are shown in Figure 11.

**Semi-SL**  Results are shown in Figure 12.

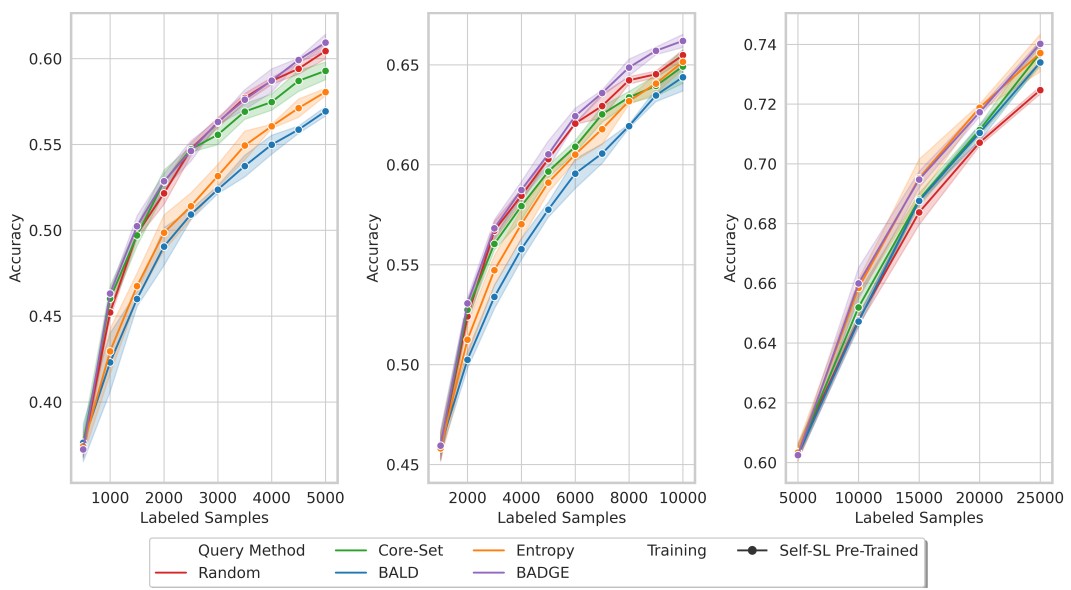

Figure 11: CIFAR-100 Self-SL

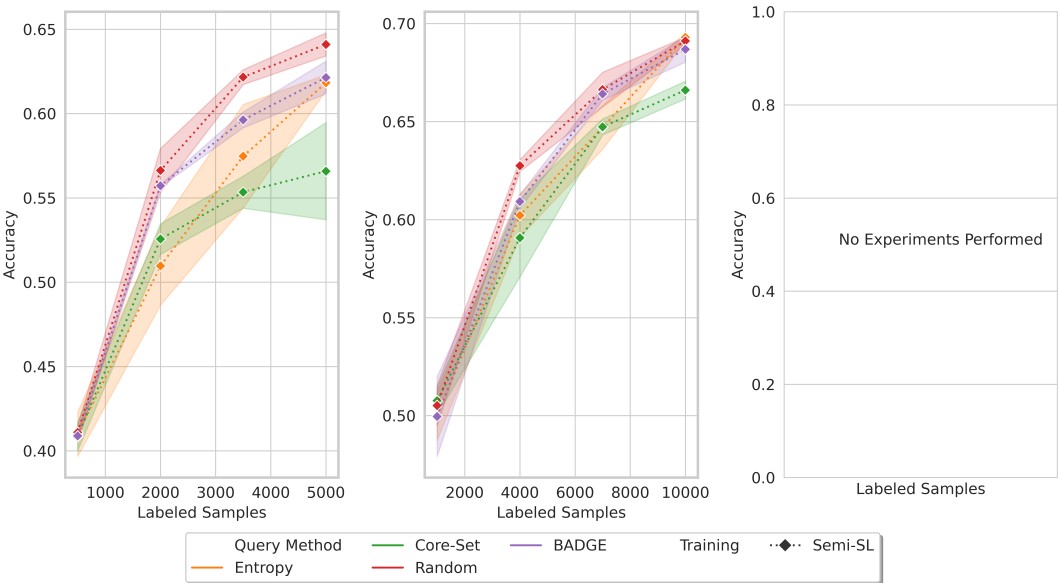

Figure 12: CIFAR-100 Semi-SL

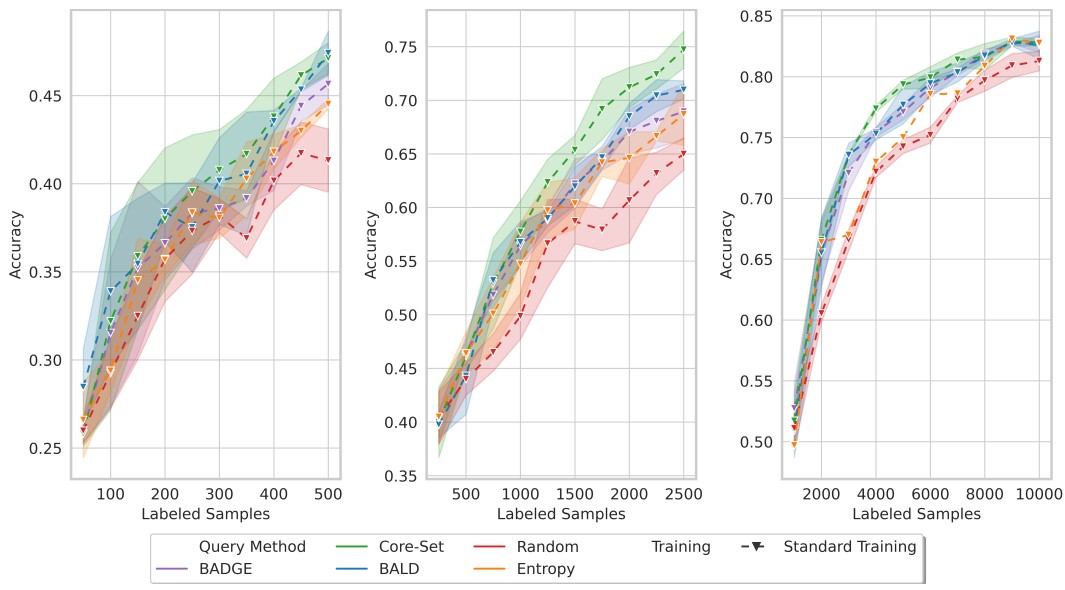

Figure 13: CIFAR-10 LT ST

### F.1.3 CIFAR-10 LT

The AUBC values are shown in Tab. 12.

Table 12: Area Under Budget Curve values for CIFAR-10 LT.

| Training | Query Method | Low-Label Mean | Low-Label STD | Medium-Label Mean | Medium-Label STD | High-Label Mean | High-Label STD |
|---|---|---|---|---|---|---|---|
| ST | BADGE | 0.3788 | 0.0152 | 0.5887 | 0.0098 | 0.7577 | 0.0004 |
| | BALD | 0.3919 | 0.0111 | 0.5935 | 0.0141 | 0.7590 | 0.0034 |
| | Entropy | 0.3740 | 0.0064 | 0.5793 | 0.0114 | 0.7454 | 0.0023 |
| | Core-Set | 0.3939 | 0.0249 | 0.6162 | 0.0200 | 0.7667 | 0.0058 |
| | Random | 0.3615 | 0.0118 | 0.5446 | 0.0214 | 0.7263 | 0.0044 |
| Self-SL | BADGE | 0.5373 | 0.0233 | 0.6501 | 0.0026 | 0.7704 | 0.0093 |
| | BALD | 0.5431 | 0.0202 | 0.6549 | 0.0097 | 0.7742 | 0.0036 |
| | Entropy | 0.5282 | 0.0225 | 0.6450 | 0.0090 | 0.7707 | 0.0061 |
| | Core-Set | 0.5298 | 0.0182 | 0.6171 | 0.0154 | 0.7555 | 0.0032 |
| | Random | 0.5397 | 0.0208 | 0.6173 | 0.0097 | 0.7554 | 0.0069 |
| Semi-SL | BADGE | 0.7233 | 0.0166 | 0.7616 | 0.0087 | – | – |
| | Entropy | 0.6934 | 0.0289 | 0.7590 | 0.0101 | – | – |
| | Core-Set | 0.6825 | 0.0103 | 0.7608 | 0.0108 | – | – |
| | Random | 0.6965 | 0.0264 | 0.7363 | 0.0077 | – | – |

**ST** Results are shown in Figure 13.

**Self-SL** Results are shown in Figure 14.

**Semi-SL** Results are shown in Figure 15.

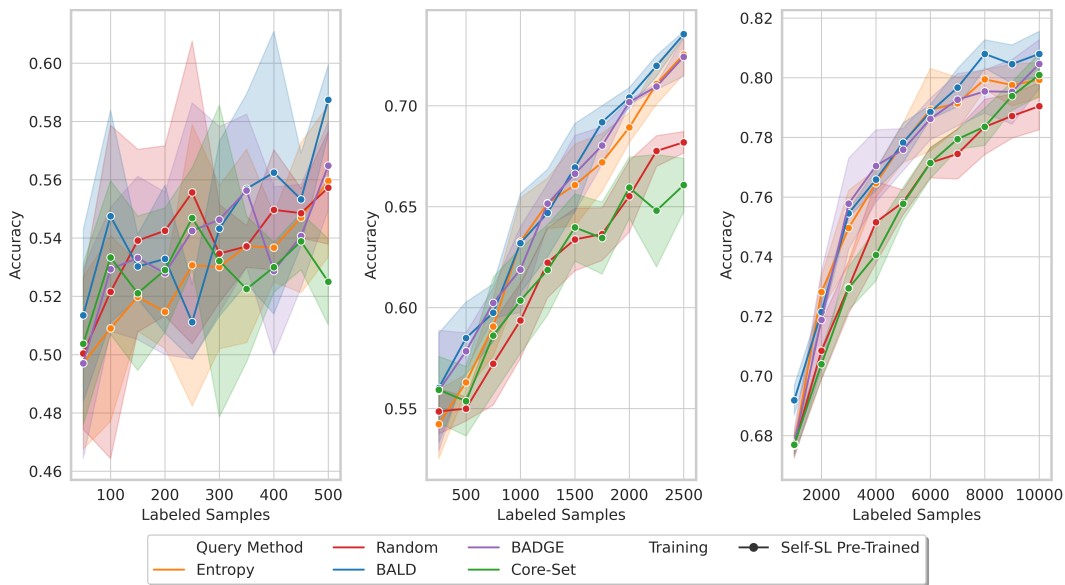

Figure 14: CIFAR-10 LT Self-SL

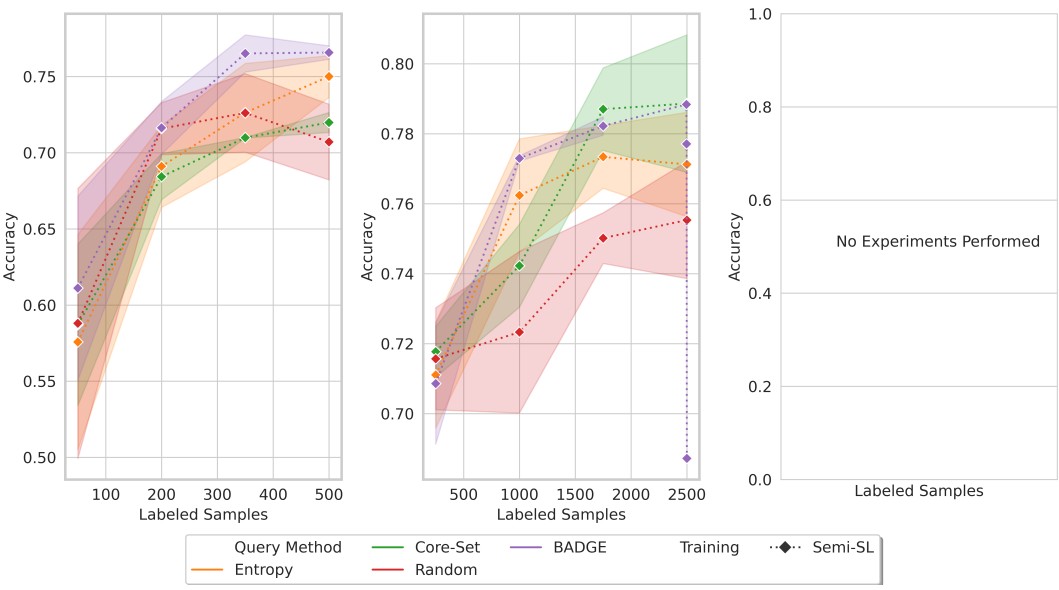

Figure 15: CIFAR-10 LT Semi-SL

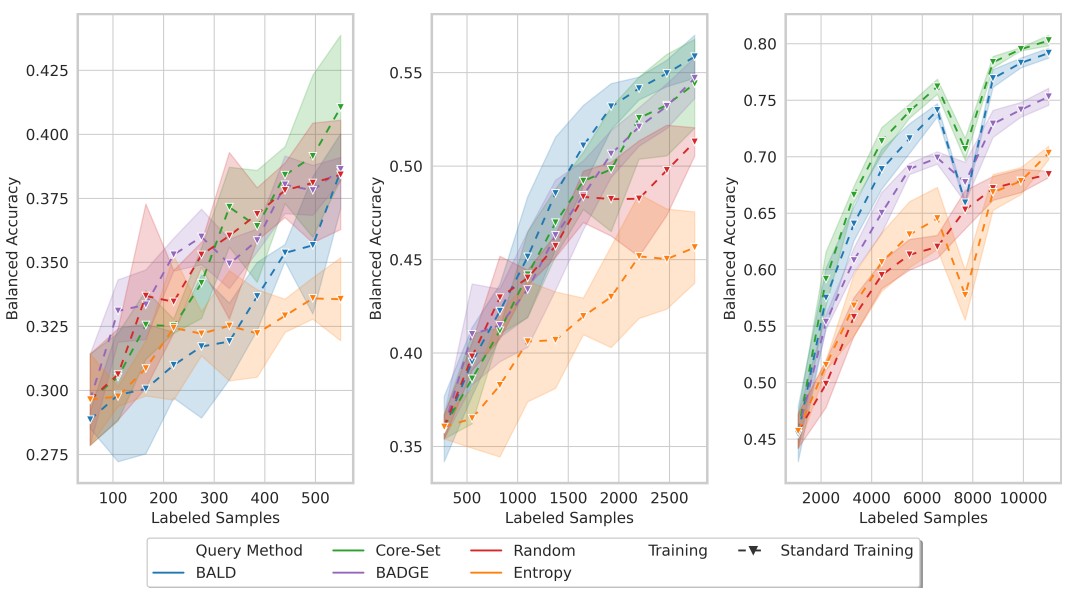

Figure 16: MIO-TCD ST

### F.1.4 MIO-TCD

The AUBC values are shown in Tab. 13.

Table 13: Area Under Budget Curve values for MIO-TCD.

|          | Label Regime | Low-Label | | Medium-Label | | High-Label | |
|----------|--------------|-----------|--------|--------------|--------|------------|--------|
|          |              | Mean | STD | Mean | STD | Mean | STD |
| Training | Query Method |      |     |      |     |      |     |
| ST       | BADGE    | 0.3539 | 0.0041 | 0.4688 | 0.0153 | 0.6614 | 0.0080 |
|          | BALD     | 0.3254 | 0.0155 | 0.4830 | 0.0092 | 0.6884 | 0.0104 |
|          | Entropy  | 0.3201 | 0.0097 | 0.4134 | 0.0230 | 0.6078 | 0.0176 |
|          | Core-Set | 0.3514 | 0.0134 | 0.4678 | 0.0181 | 0.7098 | 0.0056 |
|          | Random   | 0.3510 | 0.0151 | 0.4564 | 0.0140 | 0.6065 | 0.0120 |
| Self-SL  | BADGE    | 0.5446 | 0.0122 | 0.6365 | 0.0054 | 0.7174 | 0.0040 |
|          | BALD     | 0.4494 | 0.0102 | 0.5741 | 0.0138 | 0.6041 | 0.0092 |
|          | Entropy  | 0.5105 | 0.0092 | 0.6416 | 0.0075 | 0.6972 | 0.0029 |
|          | Core-Set | 0.5060 | 0.0082 | 0.5900 | 0.0190 | 0.6699 | 0.0166 |
|          | Random   | 0.5298 | 0.0109 | 0.6124 | 0.0032 | 0.6975 | 0.0054 |

**ST** Results are shown in Figure 16.

**Self-SL** Results are shown in Figure 17.

**Semi-SL** We performed no AL experiments due to the bad performance of Semi-SL on the starting budgets. More information can be found in Appendix F.4.

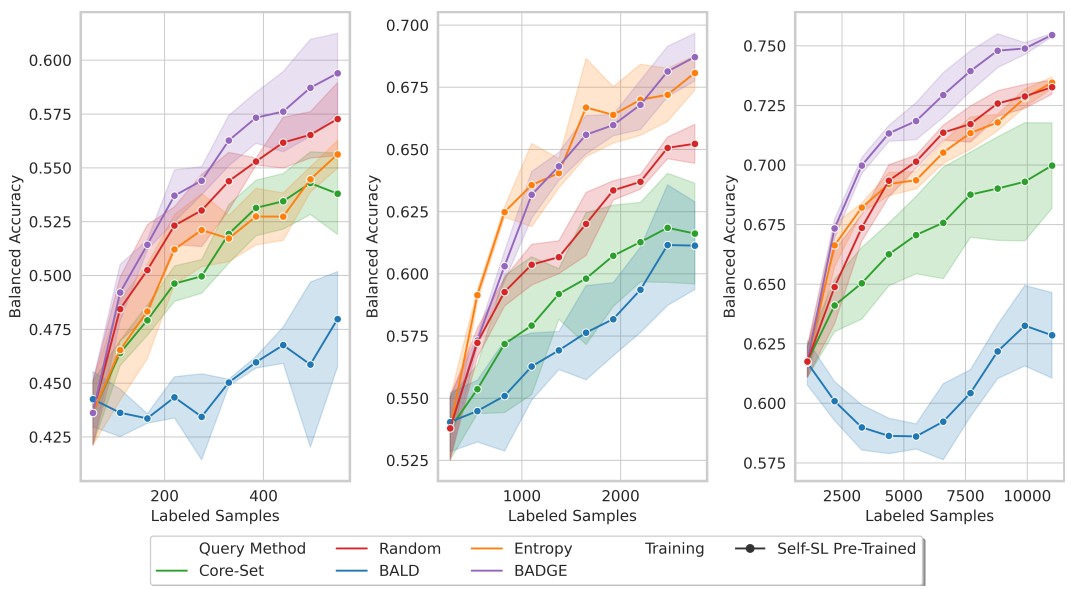

Figure 17: MIO-TCD Self-SL

### F.1.5 ISIC-2019

The AUBC values are shown in Tab. 14.

Table 14: Area Under Budget Curve values for ISIC-2019.

| | Label Regime | Low-Label | | Medium-Label | | High-Label | |
| | | Mean | STD | Mean | STD | Mean | STD |
| Training | Query Method | | | | | | |
|---|---|---|---|---|---|---|---|
| ST | BADGE | 0.3204 | 0.0099 | 0.4331 | 0.0146 | 0.5628 | 0.0101 |
| | BALD | 0.3190 | 0.0133 | 0.4521 | 0.0211 | 0.5534 | 0.0052 |
| | Entropy | 0.3241 | 0.0067 | 0.4207 | 0.0335 | 0.5631 | 0.0061 |
| | Core-Set | 0.3426 | 0.0099 | 0.4501 | 0.0139 | 0.5708 | 0.0096 |
| | Random | 0.3376 | 0.0243 | 0.4116 | 0.0201 | 0.5273 | 0.0048 |
| Self-SL | BADGE | 0.3809 | 0.0168 | 0.4679 | 0.0174 | 0.5761 | 0.0063 |
| | BALD | 0.3949 | 0.0209 | 0.4847 | 0.0018 | 0.5914 | 0.0080 |
| | Entropy | 0.3666 | 0.0165 | 0.4659 | 0.0104 | 0.5872 | 0.0096 |
| | Core-Set | 0.3752 | 0.0205 | 0.4472 | 0.0071 | 0.5556 | 0.0069 |
| | Random | 0.3736 | 0.0092 | 0.4555 | 0.0053 | 0.5547 | 0.0066 |

**ST** Results are shown in Figure 18.

**Self-SL** Results are shown in Figure 19.

**Semi-SL** We performed no AL experiments due to the bad performance of Semi-SL on the starting budgets. More information can be found in Appendix F.4.

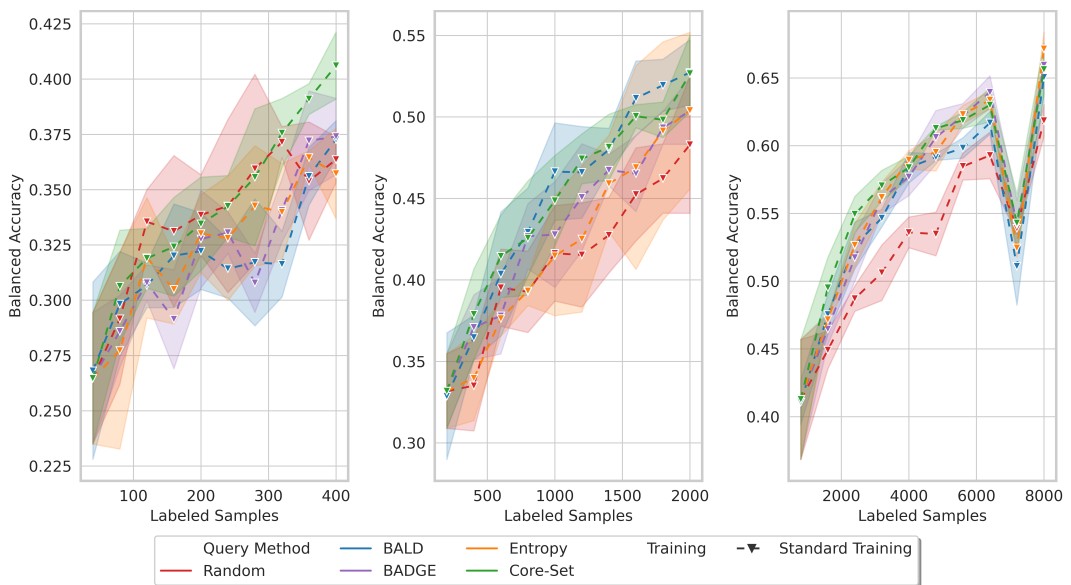

Figure 18: ISIC-2019 ST

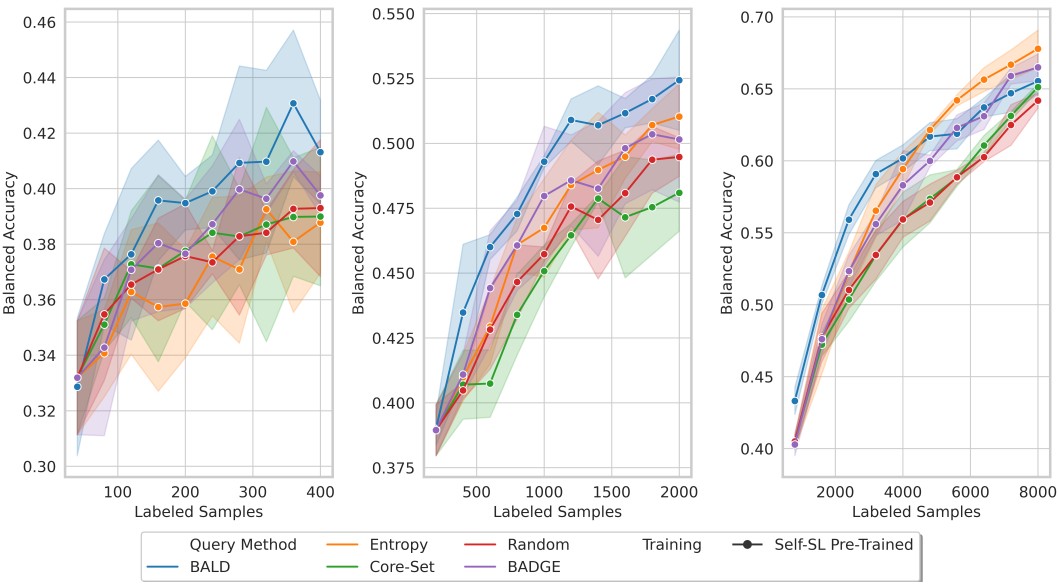

Figure 19: ISIC-2019 Self-SL Pre-Trained Models

## F.2 Low-Label Query Size

To investigate the effect of query size in the low-label regime, we conduct an ablation with Self-SL pre-trained models on CIFAR-100 and ISIC-2019. For CIFAR-100 the query sizes are 50, 500 and 2000, while for ISIC-2019 they are 10, 40 and 160.

Here the accuracies for the overlapping labeled samples are shown which are analyzed using a t-test.

**CIFAR100**   The results are shown in Tab. 15 and Tab. 16. When comparing the performance of the same QM using different query sizes only BALD and Core-Set lead to statistically significant difference in performance. While it is consistent across both comparisons for BALD with the the performance difference widening for larger labeled sets and more iterations the trend for Core-Set is not as clear.

**ISIC-2019**   The results are shown in Tab. 17 and Tab. 18. Entropy is the only QM showing a significant difference, indicating that a query size of 40 outperforms a query size of 10 for a training set size of 160. However, this behavior does not extend to the other training set sizes. Whereas, for BALD, there is a consistent trend that smaller query sizes lead to increased performance.

Table 15: Accuracies % for the low-label comparison for CIFAR100 with query sizes 50 and 500 at overlapping training set sizes. Reported as mean (std). Values with a significant difference (t-test) across query sizes are denoted with $*$.

| Labeled Samples | 1000 | | 1500 | |
|---|---|---|---|---|
| Query Size | 50 | 500 | 50 | 500 |
| BADGE | 45.75 (0.75) | 46.32 (0.20) | 50.02 (0.92) | 50.24 (0.61) |
| BALD | 45.54 (0.20) | 42.31 (1.71) | 49.41 (0.39)* | 46.00 (0.44)* |
| Core-Set | 46.53 (0.76) | 46.02 (0.74) | 49.34 (0.72) | 49.70 (0.60) |
| Entropy | 43.82 (0.40) | 42.95 (1.32) | 47.05 (0.91) | 46.75 (0.76) |
| Random | 46.10 (0.41) | 45.22 (0.34) | 49.88 (0.26) | 49.79 (0.27) |

Table 16: Accuracies % for the low-label comparison for CIFAR100 with query sizes 500 and 2000 at overlapping training set sizes. Reported as mean (std). Values with a significant difference (t-test) across query sizes are denoted with $*$.

| Labeled Samples | 2500 | | 4500 | |
|---|---|---|---|---|
| Query Size | 500 | 2000 | 500 | 2000 |
| BADGE | 54.62 (0.60) | 55.03 (0.38) | 50.92 (0.12) | 59.98 (0.60) |
| BALD | 50.92 (0.30) | 49.96 (1.43) | 55.86 (0.21)* | 52.14 (1.36)* |
| Core-Set | 54.72 (0.16)* | 53.52 (0.33)* | 58.71 (0.61) | 58.28 (0.43) |
| Entropy | 51.41 (0.53) | 51.79 (0.76) | 57.12 (0.53) | 57.53 (0.43) |
| Random | 54.71 (0.38) | 54.38 (0.21) | 59.98 (0.60) | 59.48 (0.48) |

Table 17: Accuracies % for the low-label comparison for ISIC-2019 with query sizes 10 and 40 at overlapping training set sizes. Reported as mean (std). Values with a significant difference (t-test) across query sizes are denoted with $*$.

| Labeled Sample | 80 | | 120 | | 160 | | 200 | | 240 | |
|---|---|---|---|---|---|---|---|---|---|---|
| Query Size | 10 | 40 | 10 | 40 | 10 | 40 | 10 | 40 | 10 | 40 |
| BADGE | 34.39 (0.86) | 36.03 (0.39) | 37.58 (1.93) | 36.40 (1.10) | 37.58 (2.51) | 37.24 (2.15) | 38.32 (1.59) | 37.52 (1.16) | 38.78 (1.64) | 38.94 (2.97) |
| BALD | 38.11 (1.42) | 36.51 (1.26) | 38.80 (1.19) | 37.26 (4.90) | 40.40 (1.76) | 39.23(1.89) | 42.15 (1.49) | 40.09 (2.59) | 42.26 (0.97) | 40.18 (2.70) |
| Core-Set | 35.57 (1.49) | 35.38 (0.50) | 36.87 (1.73) | 36.52 (2.24) | 38.55 (2.37) | 38.19 (1.69) | 38.41 (1.80) | 37.19 (2.93) | 39.04 (2.94) | 38.30 (1.66) |
| Entropy | 35.19 (0.63) | 34.87 (0.91) | 37.06 (1.63) | 38.59 (2.06) | 36.95 (1.13)* | 39.67 (0.55)* | 38.25 (3.11) | 39.93 (2.38) | 39.38 (2.62) | 40.26 (2.49) |
| Random | 36.10 (0.70) | 35.69 (0.68) | 37.57 (1.79) | 37.65 (2.15) | 37.43 (1.77) | 40.62 (0.62) | 38.41 (1.46) | 40.29 (1.78) | 3835 (2.43) | 41.14 (0.37) |

Table 18: Accuracies % for the low-label comparison for ISIC-2019 with query sizes 40 and 160 at overlapping training set sizes. Reported as mean (std). Values with a significant difference (t-test) across query sizes are denoted with ∗.

| Labeled Samples | 200 | | 360 | |
|---|---|---|---|---|
| Query Size | 40 | 160 | 40 | 160 |
| BADGE | 37.52 (1.16) | 38.11 (1.07) | 42.90 (3.29) | 42.86 (1.44) |
| BALD | 40.09 (2.59) | 39.67 (2.54) | 44.49 (1.78) | 42.83 (1.05) |
| Core-Set | 37.79 (2.93) | 37.71 (3.18) | 39.56 (1.48) | 39.47 (1.04) |
| Entropy | 39.93 (2.38) | 37.80 (1.46) | 42.03 (1.16) | 40.82 (1.46) |
| Random | 40.29 (1.78) | 40.80 (2.34) | 42.10 (2.29) | 42.10 (2.29) |

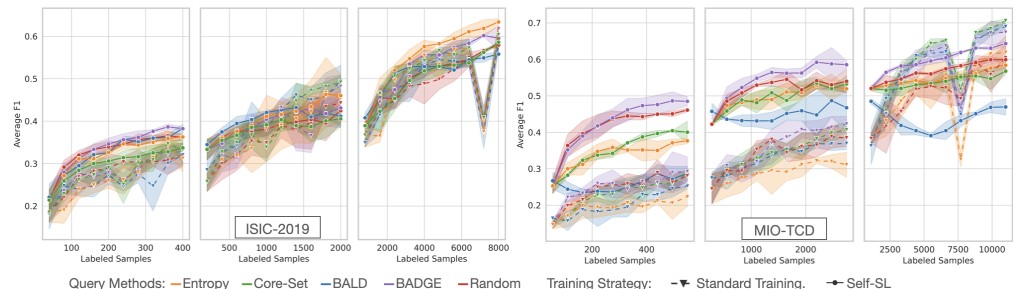

Figure 20: Macro-averaged F1-scores for ISIC-2019 and MIO-TCD datasets. Across the board, BADGE is still the best-performing QM which can also be seen in Table 19. Please interpret the results with care, as the model configurations are optimized for balanced accuracy.

Table 19: Area Under Budget Curve values based on macro-averaged F1-scores for ISIC-2019 and MIO-TCD (corresponding plots in Figure 20).

(a) ISIC-2019

| Training | Label Regime Query Method | Low-Label Mean | STD | Mid-Label Mean | STD | High-Label Mean | STD |
|---|---|---|---|---|---|---|---|
| Standard Training | BADGE | 0.2791 | 0.0081 | 0.3810 | 0.0167 | 0.5059 | 0.0054 |
| | BALD | 0.2587 | 0.0196 | 0.4050 | 0.0216 | 0.4866 | 0.0100 |
| | Entropy | 0.2652 | 0.0147 | 0.3763 | 0.0453 | 0.4988 | 0.0056 |
| | Core-Set | 0.2800 | 0.0105 | 0.3937 | 0.0082 | 0.5139 | 0.0141 |
| | Random | 0.2817 | 0.0153 | 0.3638 | 0.0292 | 0.4774 | 0.0035 |
| Self-SL Pre-Trained | BADGE | 0.3390 | 0.0107 | 0.3978 | 0.0053 | 0.5380 | 0.0067 |
| | BALD | 0.3273 | 0.0054 | 0.4065 | 0.0047 | 0.5173 | 0.0008 |
| | Entropy | 0.3232 | 0.0039 | 0.4081 | 0.0061 | 0.5586 | 0.0143 |
| | Core-Set | 0.3010 | 0.0129 | 0.3833 | 0.0094 | 0.5045 | 0.0123 |
| | Random | 0.3333 | 0.0033 | 0.3852 | 0.0153 | 0.5100 | 0.0059 |

(b) MIO-TCD

| Training | Label Regime Query Method | Low-Label Mean | STD | Mid-Label Mean | STD | High-Label Mean | STD |
|---|---|---|---|---|---|---|---|
| Standard Training | BADGE | 0.2503 | 0.0111 | 0.3604 | 0.0249 | 0.5759 | 0.0075 |
| | BALD | 0.2023 | 0.0127 | 0.3325 | 0.0082 | 0.5758 | 0.0116 |
| | Entropy | 0.1957 | 0.0208 | 0.2929 | 0.0218 | 0.5066 | 0.0290 |
| | Core-Set | 0.2301 | 0.0155 | 0.3450 | 0.0203 | 0.5945 | 0.0099 |
| | Random | 0.2446 | 0.0091 | 0.3438 | 0.0132 | 0.5085 | 0.0093 |
| Self-SL Pre-Trained | BADGE | 0.4295 | 0.0205 | 0.5487 | 0.0129 | 0.5994 | 0.0035 |
| | BALD | 0.2560 | 0.0132 | 0.4488 | 0.0236 | 0.4335 | 0.0088 |
| | Entropy | 0.3398 | 0.0104 | 0.4977 | 0.0095 | 0.5481 | 0.0078 |
| | Core-Set | 0.3525 | 0.0066 | 0.4970 | 0.0160 | 0.5375 | 0.0138 |
| | Random | 0.4175 | 0.0158 | 0.5194 | 0.0101 | 0.5681 | 0.0036 |

## F.3 Macro averaged F1-scores

Additionally, we provide the macro-averaged F1-scores for ISIC-2019 and MIO-TCD dataset. A plot showing the performance of both ST and Self-SL models is shown in Figure 20 and the resulting AUBC values are shown in Tab. 19.

## F.4 Semi-Supervised Learning

Results of FixMatch for all HPs on the whole validation splits are shown separately for MIO-TCD in Tab. 20 and ISIC-2019 in Tab. 21. Based on the performance which did not improve substantially over even ST models we decided to omit all further AL experiments.

Table 20: MIO-TCD FixMatch results reported on the test sets (balanced accuracy in %). Reported as mean (std).

| FixMatch Sweep MIO-TCD Labeled Train Samples | Learning Rate | Weight Decay | Balanced Accuracy (Test) |
|---|---|---|---|
| 55 | 0.3 | 5E-3 | 18.1(1.1) |
| | | 5E-4 | 28.0(0.4) |
| | 0.03 | 5E-3 | 26.8(0.5) |
| | | 5E-4 | 29.7(2.4) |
| 275 | 0.3 | 5E-3 | 20.2(6.2) |
| | | 5E-4 | 28.2(1.5) |
| | 0.03 | 5E-3 | 31.4(0.7) |
| | | 5E-4 | 36.0(1.8) |

Table 21: ISIC-2019 FixMatch results reported on the test sets (balanced accuracy in %). Reported as mean (std).

| FixMatch Sweep ISIC-2019 | | | |
|---|---|---|---|
| Labeled Train Samples | Learning Rate | Weight Decay | Balanced Accuracy (Test) |
| 40 | 0.3 | 5E-3 | 14.0(2.5) |
| | | 5E-4 | 26.9(1.5) |
| | 0.03 | 5E-3 | 24.5(4.4) |
| | | 5E-4 | 31.3(1.7) |
| 200 | 0.3 | 5E-3 | 15.2(3.6) |
| | | 5E-4 | 19.1(1.5) |
| | 0.03 | 5E-3 | 26.3(0.8) |
| | | 5E-4 | 24.3(3.6) |

# G    Discussion and further observations

The results are interpreted based on the assumption that a QM performing on a similar level as Random is not a drawback as long as it brings in other settings performance improvements over random queries. This mostly follows in line with the PPM as a performance metric but mostly focuses on the row that compares each QM with random queries. However, if a QM shows behavior leading to much worse behavior than random as Entropy does or shows signs of the cold start problem, we deem this as highly problematic. In these settings, one loses significant performance whilst paying a cost in computing and setup corresponding to AL. Therefore, we use random queries as a baseline for all QMs.
Based on this our recommendation for BADGE is given for Self-SL and ST trainings.
The main disadvantage of this approach is that absolute performance difference are not captured in this aggregated format.

## H   Comparing random-sampling baselines across studies

Here we compare the performance of random-sampling baselines on the most commonly utilized dataset CIFAR-10 and CIFAR-100 across different studies for ST, Self-SL and Semi-SL models along strategic point where overlap in between papers occurs. For CIFAR-10 the results of this comparison are shown for the high-label regime in Tab. 22 and the low- and mid-label regime in Tab. 23. Similarly for CIFAR-100 the results are shown in Tab. 24 for the high-label regime and Tab. 25 for the low- and mid-label regimes. Overall our ST random baselines outperform all other random baselines. Our Self-SL models also outperform the only other relevant literature [6] on CIFAR-10. Further, our Semi-SL models also outperform the relevant literature [20, 43] on CIFAR-10 and CIFAR-100.

Table 22: Comparison of random baseline model accuracy in % on the test set for the high label-regime for CIFAR-10 across different papers. Best performing models for each training strategy are **highlighted**. Values denoted with – represent not performed experiments. Values with a denoted with a * are reprinted from [44]. Values which are sourced from a graph are subject to human read-out error.

| Information | | | | Number Labeled Training Samples | | | | | |
|---|---|---|---|---|---|---|---|---|---|
| Paper | Training | Model | Source | 1k | 2k | 5k | 10k | 15k | 20k |
| QBC | ST | DenseNet121 | Graph | | | 74* | 82.5* | - | - |
| VAAL | ST | VGG16 | Graph | - | - | 61.35* | 68.17* | 72.96* | 75.99* |
| CoreSet | ST | VGG16 | Graph | - | - | 60* | 68* | 71* | 74* |
| Agarwal et al. | ST | VGG16 | Graph | - | - | 61.5 | 68 | 72 | 76 |
| Munjal-SR | ST | VGG16 | Table | - | - | 82.16 | 85.07 | 89.43 | 91.16 |
| Mittal et al. | ST | WRN28-2 | Graph | 57 | 73 | 82.5 | 86 | 90.7 | 92 |
| LLAL | ST | ResNet18 | Graph | 51 | 63 | 81* | 87* | - | - |
| CoreCGN | ST | ResNet18 | Graph | 50 | 64 | 80* | 85.5* | - | - |
| TA-VAAL | ST | ResNet18 | Graph | 50 | 65 | 81* | 87.5* | - | - |
| Krishnan et al. | ST | ResNet18 | Graph | 47 | 60 | 78 | 86 | - | - |
| Yi et al. | ST | ResNet18 | Graph | 47.5 | 56 | 78 | 86 | - | - |
| Bengar et al. | ST | ResNet18 | Graph | 45 | 55 | 73 | 81 | 85 | 88 |
| Beck et al. | ST | ResNet18 | Graph | 55 | - | - | 84 | 85 | 90.5 |
| Zhan et al. | ST | ResNet18 | Graph | 45 | - | - | 76 | - | - |
| Munjal-SR | ST | ResNet18 | Table | - | - | 84.69 | 88.45 | 89.98 | 92.29 |
| Ours | ST | ResNet18 | Table | **72.4** | **79.8** | **85.5** | **90.5** | - | - |
| Bengar et al. | Self-SL | ResNet18 | Graph | **87** | 88 | 89.5 | 90.5. | 91 | 91.5 |
| Ours | Self-SL | ResNet18 | Table | 86.2 | **88.3** | **90.1** | **91.4** | - | - |
| Mittal et al. | Semi-SL | WRN28-2 | Graph | 88 | 91 | 92.5 | 93.8 | 94 | 94.5 |
| Gao et al. | Semi-SL | WRN28-2 | Graph | 91.5 | 91 | - | - | - | - |
| Ours | Semi-SL | ResNet18 | Table | **94.7** | **95.0** | - | - | - | - |

Table 23: Comparison of random baseline model accuracy in % on the test set for the low- and mid-label regime for CIFAR-10 across different papers. Best performing models for each training strategy are **highlighted**. Values denoted with – represent not performed experiments. Values which are sourced from a graph are subject to human read-out error.

| Information Paper | Training | Model | Source | Number Labeled Training Samples 50 | 100 | 200 | 250 | 500 |
|---|---|---|---|---|---|---|---|---|
| Chan et al. | ST | WRN28-2 | Table | - | - | - | 40.9 | - |
| Mittal et al. | ST | WRN28-2 | Graph | - | - | - | 36 | 48 |
| Bengar et al. | ST | ResNet18 | Graph | - | - | - | - | 38 |
| Ours | ST | ResNet18 | Table | **25.1** | **32.3** | **44.4** | **47.0** | **61.2** |
| Chan et al. | Self-SL | WRN28-2 | Table | - | - | - | 76.7 | - |
| Bengar et al. | Self-SL | ResNet18 | Graph | 62 | **77** | 81 | 83 | 85 |
| Ours | Self-SL | ResNet18 | Table | **71.3** | 76.8 | **81.2** | 81.4 | 84.1 |
| Chan et al. | Semi-SL | WRN28-2 | Table | - | - | - | **93.1** | - |
| Mittal et al. | Semi-SL | WRN28-2 | Graph | - | - | - | 82 | 85 |
| Gao et al. | Semi-SL | WRN28-2 | Table | - | 47.9 | 89.2 | 90.2 | - |
| Ours | Semi-SL | ResNet18 | Graph | **90** | **91** | **93** | 93 | **94** |

Table 24: Comparison of random baseline model accuracy in % on the test set for the high-label regime for CIFAR-100 across different papers. Best performing models for each training strategy are **highlighted**. Values denoted with – represent not performed experiments. Values which are sourced from a graph are subject to human read-out error.

| Information Paper | Training | Model | Source | Number Labeled Training Samples 5k | 10k | 15k | 20k |
|---|---|---|---|---|---|---|---|
| Agarwal et al. | ST | VGG16 | Graph | 28 | 35 | 41.5 | 46 |
| Agarwal et al. | ST | ResNet18 | Graph | 29.5 | 38 | 45 | 49 |
| Core-Set | ST | VGG16 | Graph | 27 | 37 | 42 | 49 |
| VAAL | ST | VGG16 | Graph | 28 | 35 | 42 | 46 |
| Munjal et al. | ST | VGG16 | Graph | 39.44 | 49 | 55 | 59 |
| VAAL | ST | ResNet18 | Graph | 28 | 38 | 45 | 49 |
| TA-VAAL | ST | ResNet18 | Graph | 43 | 52 | 60 | 63.5 |
| Bengar et al. | ST | ResNet18 | Graph | 27 | 45 | 52 | 58 |
| Beck et al. | ST | ResNet18 | Graph | 40 | 53 | 60 | 64 |
| Zhan et al. | ST | ResNet18 | Graph | - | 39 | - | - |
| Munjal et al. | ST | ResNet18 | Table | ? | 61.1 | **66.9** | 69.8 |
| Mittal et al. | ST | WRN28-2 | Graph | 44.9 | 58 | 64 | 68 |
| Ours | ST | ResNet18 | Table | **49.2** | **61.3** | 66.7 | **70.2** |
| Bengar et al. | Self-SL | ResNet18 | Table | 60 | 63 | 63.5 | 64 |
| Ours | Self-SL | ResNet18 | Table | **60.4** | **64.8** | **68.4** | **70.7** |
| Mittal et al. | Semi-SL | WRN28-2 | Graph | 59 | 65 | 70 | 71 |
| Gao et al. | Semi-SL | WRN28-2 | Table | **63.4** | 67 | 68 | 70 |
| Ours | Semi-SL | ResNet18 | Graph | **63.5** | **68.5** | - | - |

Table 25: Comparison of random baseline model accuracy in % on the test set for the low- and mid- label regime for CIFAR-100 across different papers. Best performing models for each training strategy are **highlighted**. Values denoted with – represent not performed experiments. Values which are sourced from a graph are subject to human read-out error.

| Information | | | | Number Labeled Training Samples | | | |
|---|---|---|---|---|---|---|---|
| Paper | Training | Model | Source | 500 | 1000 | 2000 | 2500 |
| Chan et al. | ST | WRN28-2 | Table | - | - | - | 33.2 |
| Mittal et al. | ST | WRN28-2 | Graph | 9 | 12 | 24 | 27 |
| TA-VAAL | ST | ResNet18 | Graph | - | - | 20 | - |
| Bengar et al. | ST | ResNet18 | Graph | 9 | 12 | 17 | - |
| Ours | ST | ResNet18 | Table | **14.0** | **22.4** | **32.0** | **36.3** |
| Chan et al. | Self-SL | WRN28-2 | Table | - | - | - | 49.1 |
| Bengar et al. | Self-SL | ResNet18 | Table | **47** | **50** | **56** | - |
| Ours | Self-SL | ResNet18 | Table | 37.3 | 45.2 | 52.2 | 54.7 |
| Chan et al. | Semi-SL | WRN28-2 | Table | - | - | - | **67.6** |
| Mittal et al. | Semi-SL | WRN28-2 | Graph | 26 | 35.5 | 44.5 | 49 |
| Ours | Semi-SL | ResNet18 | Graph | **41** | - | **56.5** | - |

# I  Detailed limitations

Additionally to the limitations already discussed in Sec. 4 we would like to critically reflect on the following points:

**Query methods** We only evaluate four different QMs which is only a small sub-selection of all the QMs proposed in the literature. We argue that this may not be optimal, however, deem it justified due to the variety of other factors which we evaluated. Further, we excluded all QMs which induce changes in the classifier (s.a. LLAL [55]) or add a substantial additional computational cost by training new components (s.a. VAAL [51]). These QMs might induce changes in the HPs for every dataset and were therefore deemed too costly to properly optimize.
We leave a combination of P4 with these QMs for future research.

**Validation set size** The potential shortcomings of our validation set were already discussed. However, we would like to point out that a principled inclusion of K-Fold Cross-Validation into AL might alleviate this problem. This would also give direct access to ensembles which have been shown numerous times to be beneficial with regard to final performance (also in AL) [5]. How this would allow us to assess performance gains in practice and also make use of improved techniques for performance evaluation s.a. Active Testing [34] in the same way as our proposed solution shown in Figure 4 is not clear to us. Therefore we leave this point up for future research.

**Performance of ST models** On the imbalanced datasets, the performance of our models is not steadily increasing for more samples which can be traced back to sub-optimal HP selection according to [44]. We believe that our approach of simplified HP tuning improves over the state-of-the-art in AL showcased by the superior performance of our models on CIFAR-10 and CIFAR-100. However, regularly re-optimizing HPs might be an alternative solution.

**Performance of Self-SL models** Our Self-SL models are outperformed on the low-label regime on CIFAR-100 by the Self-SL models by [6], whereas on the medium- and high-label regime our Self-SL models outperform them. We believe that this might be due to our fine-tuning schedule and the possibility that Sim-Siam improves over SimCLR on CIFAR-100. Since our Self-Sl models still outperform most Semi-SL models in the literature we believe that drawing conclusions from our results is still feasible. An interesting research direction would be to make better use of the Self-SL representations s.a. improved fine-tuning regimes [37].

**No Bayesian Query Methods for Semi-SL** The Semi-SL models were neither combined with BALD nor BatchBALD as query functions, even though we showed that small query sizes and BatchBALD can counteract the cold-start problem. Further our Semi-SL models had bigger query sizes by a factor of three, possibly additionally hindering performance gains obtainable with AL. However, in previous experiments with FixMatch, we were not able to combine it with Dropout whilst keeping the performance of models without dropout. This clearly might have been an oversight by us, but we would like to point out that in the works focusing on AL, using Semi-SL without bayesian QMs is common practice [20, 43]

**Changing both starting budget and query size** We correlated the two parameters (smaller query size for smaller starting budget etc.) since 1) in practice, we deem the size of the starting budget to be dependent on labeling cost (therefore, large query sizes for small starting budgets are unrealistic and vice versa) and 2) In this work, we are especially interested in smaller starting budgets ("cold-start" territory) compared to the ones in the literature, since AL typically shows robust performance for larger starting budgets. Theory shows that our adapted smaller query size for this case can only positively affect the result [19, 32]. The only possible confounder could be that we interpret the performance of a small starting budget too positively due to a hidden effect of the smaller query size. However, we performed the low-label query size ablation, showcasing that varying the query size for small starting budgets did not have considerable effects on performance for all QMs, except BALD, where, a clear performance increase for smaller query sizes was observed.

## I.1  Instability of hyperparameters for class imbalanced datasets

The substantial dip in performance on MIO-TCD and ISIC-2019 for approx 7k samples shown in Figure 16 and Figure 18 is ablated in Tab. 26 where we show that simply changing the learning rate leads to stabilizing the performance on both datasets for these cases.

Table 26: Ablation study on the performance-dip on MIO-TCD and ISIC-2019 for ST models with regard to HP. Reported as mean (std).

| Dataset | Labeled Train Set | Data Augmentation | Learning Rate | Weight Decay | Balanced Accuracy (Val) | Balanced Accuracy (Test) |
|---|---|---|---|---|---|---|
| ISIC-2019 | 7200 | RandAugmentMC (ISIC) | 0.1 | 5E-3 | 54.4(1.2) | 52.6(1.9) |
| | | | | 5E-4 | 57.2(1.7) | 55.4(0.9) |
| | | | 0.01 | 5E-3 | 58.0(1.7) | 55.6(1.0) |
| | | | | 5E-4 | 55.6(1.9) | 54.5(2.1) |
| MIO-TCD | 7700 | RandAugmentMC (ImageNet) | 0.1 | 5E-3 | 65.7(1.8) | 64.3(1.1) |
| | | | | 5E-4 | 65.9(2.4) | 63.6(2.7) |
| | | | 0.01 | 5E-3 | 64.1(1.2) | 62.9(1.0) |
| | | | | 5E-4 | 63.8(0.7) | 62.2(1.1) |

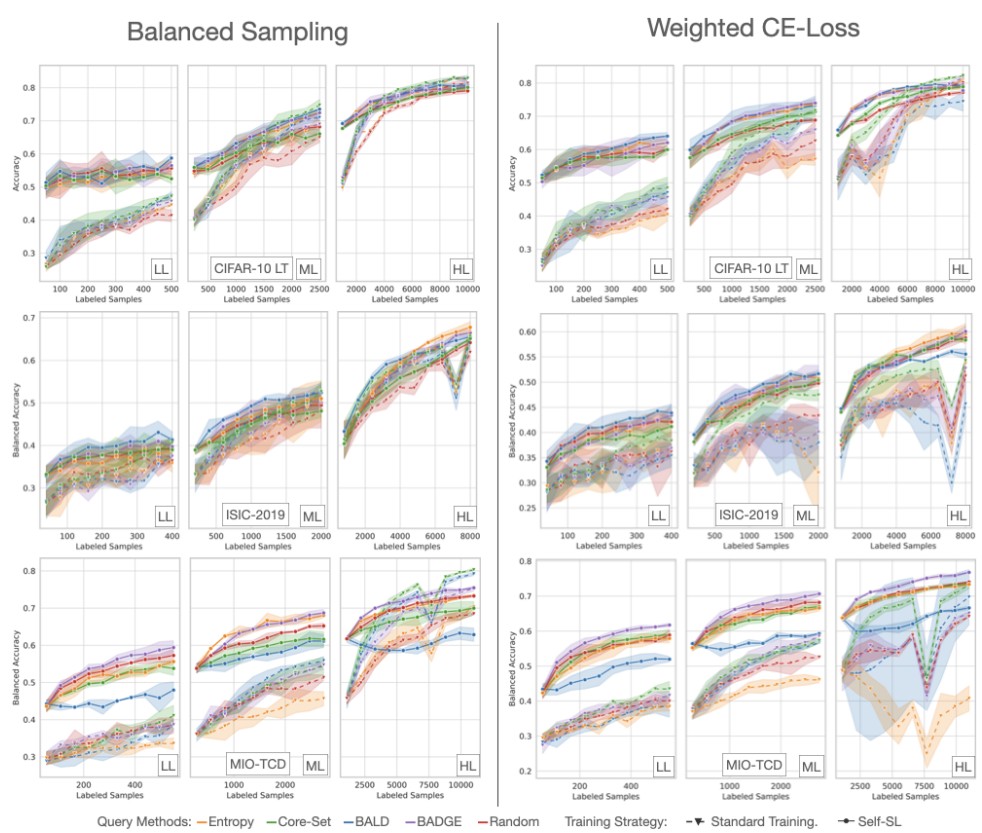

Figure 21: Comparison between balanced sampling and weighted cross-entropy-loss (weighted CE-Loss). Whereas the ST models overall seem to benefit more from balanced sampling, the Self-SL models perform slightly better for weighted CE-Loss. Generally the observed performance gains in the imbalanced settings are still present.

However, this dip in performance also arises using weighted Cross-Entropy (weighted CE-Loss) as a loss function as shown in the following ablation Figure 21.

