# OpenReview forum: "Navigating the Pitfalls of Active Learning Evaluation: A Systematic Framework for Meaningful Performance Assessment"
_NeurIPS.cc/2023/Conference — NeurIPS 2023 poster_

### Official Review · Reviewer_D3At · 2023-07-05

**Soundness:** 3 good
**Presentation:** 2 fair
**Contribution:** 3 good
**Rating:** 7
**Confidence:** 4

**Summary:**

The authors identify five main pitfalls that have impacted previous Active Learning (AL) empirical evaluations. They propose a novel evaluation framework that avoids these pitfalls, and they study both Active Learning and its potential synergies with Semi-Supervised and Self-supervised learning. The paper provides a comprehensive empirical evaluation, together with copious appendices.

**Strengths:**

The paper's main contribution is to carefully catalogue and map the various critical aspects of a proper AL evaluation, aka The Five Pitfalls; in contrast, prior work has properly addressed at most 3 of the 5 pitfalls. The thorough empirical evaluation emphasizes the proper way to create an evaluation framework for active learning, and how to also take into account Semi- & Self- Supervised methods. The paper is fairly easy to read and follow.

**Weaknesses:**

The paper can be further improved along the readability dimension.

First of all, in its current form, the paper has "too much content in too little space" Even though, at first glance, the current split between "the main paper" and "the appendices" seems reasonable, the paper would greatly benefit by compressing Section 2.3 (which could be then added "as is" in an APPENDIX). On the other hand, Section 4 is hard to follow because the graphs in Fig 2 are too small to read, so the prose is not really supported by Fig 2. One possible way to address this issue would be to move most of the 3-page Section 4 to the APPENDICES and replace it with (A) a brief, 1-page summary of the results - no figures, and (B) a new section with an INTUITIVE, PRESCRIPTIVE section that crisply summarizes the main findings: "the ideal evaluation framework for active learning for image classification is the following ..."

Second, the idea of "rolling-out Active Learning" to "unseen data sets different from the labeled development data" (first introduced at lines 32-33) is mentioned 13 times without being properly explained. In this reviewer's opinion, the idea is worth being presented in the main part of the paper, but the authors should invest the necessary 2-3 paragraphs to crisply & intuitively explain it.

OTHER COMMENTS:
- active learning does not begin/end with DL and image classification. As such, the authors could easily take the 1-2 sentences required to explicitly define the scope of their investigation.

**Questions:**

1) the four "motivating papers (lines 23-24)" [6, 19, 38, 39] use, besides CIFAR-10/100, ImageNet and Tiny ImageNet; why didn't you consider these datasets, given that some of those papers' mistaken claims are made on them?

2) on line 198-199, you claim that the three CIFARs are "developmental datasets," while the other two are used for roll-out. While this is a great idea, an even better one would be to (i) drop the CIFARs, which are too "nice & easy" to begin with, (ii) add another 2-3 non-CIFAR, real-world datasets, and (iiI) perform either Cross-Validation or Leave-One-Out on the 5 datasets - say, by taking turns in using 1-4 of them as dev datasets, and the remaining one(s) as test dataset. Did you consider such an approach?

3) I am not sure if this is just a flaw of the empirical evaluation, or is should be added as a 6th pitfall, but it is always bothersome to see that the CIFAR-10/100 evaluations of an active learner ends way earlier than the SOTA performance of 99% and 96%, respectively (in this paper, the learning curves end around 94% and 76%, respectively). After all, AL is NOT supposed to be for "quickly reaching mediocre performance," but rather "creating the best model with less data than required by random sampling."  It could be that this is due to the uniform distribution of the CIFARs (thus making them unrealistic & unsuitable for this evaluation), but this issue should be addressed head-on rather omitted.



**Limitations:**

See Weaknesses & Questions above

---

> ### Author Rebuttal · Authors · 2023-08-09
>
> We sincerely thank you for taking the time to carefully review our paper as well as for your insightful comments and questions. Note that the major changes made in our manuscript are detailed in the general reply above.
>
> -> Presentational issues: “[...] the paper has "too much content in too little space". [...]the paper would greatly benefit by compressing Section 2.3 .[...] the graphs in Fig 2 are too small to read[...]. ”
> - We thank you for these detailed suggestions. Based on these, we restructured our manuscript to improve readability and clarity. Details on this restructuring are provided in the general reply. Further, we would like to highlight that all results in Fig. 2 are also shown in larger individual figures in Appendix F, which we reference in Section 4 (line260). We made this reference more prominent in the manuscript.
>
> -> “Second, the idea of "rolling-out Active Learning" to "unseen data sets different from the labeled development data" [...] is [...] not properly explained. [...] the idea is worth being presented in the main part of the paper [...].”
> - Thank you for pointing this out. We now provide an intuitive and crips explanation for the concept of roll-out datasets. The updated descriptive part of our core concepts is detailed in the general reply.
>
> -> “active learning does not begin/end with DL and image classification. As such, the authors could easily take the 1-2 sentences required to explicitly define the scope of their investigation.”
> - We thank the author for this suggestion and added the following description of the scope in:
>     - Abstract (Line 3):  “...research on improving AL query methods for deep active classification of image data…”
>     - Section 1 (Line 40): “To this end, we present an evaluation framework for deep active classification of image data that overcomes the five pitfalls and demonstrate the relevance of this contribution by means of a large-scale empirical study spanning various datasets, query methods, AL settings, and training paradigms”.
>     - Section 2.1 as the first sentence: “The scope of our study includes deep-learning based active learning methods used for image classification.”
>
>
> -> “the four "motivating papers (lines 23-24)" [6, 19, 38, 39] use, besides CIFAR-10/100, ImageNet and Tiny ImageNet; why didn't you consider these datasets, given that some of those papers' mistaken claims are made on them?”
> - We used the CIFAR-X datasets as they were used by practically all related work so that we can compare performance across studies (shown in Appendix H) and decided to focus on real-world data on ISIC-2019 and MIO-TCD, which span a broader diversity of datasets compared to adding the suggested ones (ImageNet, CIFAR-X, STL-10). Further, Self-SL and Semi-SL methods are usually developed on these datasets; thus, using these might have biased the evaluation in their favor.
>
>
> -> “on line 198-199, you claim that the three CIFARs are "developmental datasets," while the other two are used for roll-out. While this is a great idea, an even better one would be to (i) drop the CIFARs, which are too "nice & easy" to begin with, (ii) add another 2-3 non-CIFAR, real-world datasets, and (iiI) perform either Cross-Validation or Leave-One-Out on the 5 datasets - say, by taking turns in using 1-4 of them as dev datasets, and the remaining one(s) as test dataset. Did you consider such an approach?”
> - We agree that the evaluation could be improved with additional datasets aside from the CIFARs (reason for inclusion see one paragraph above). We did not consider Cross-Validation (CV) for the evaluation and opted for a static split instead, as we (A) wanted to minimize leakage of knowledge into the test datasets and (B)  CV would increase the required compute for the experiments by a factor of at least #folds.
> The general idea, however, is very interesting and with proper execution might further improve the AL evaluation protocol.
> We added this comment to the conclusion & take-aways section in the manuscript.
>
> -> “I am not sure if this is just a flaw of the empirical evaluation, or is should be added as a 6th pitfall, but it is always bothersome to see that the CIFAR-10/100 evaluations of an active learner ends way earlier than the SOTA performance of 99% and 96%, respectively (in this paper, the learning curves end around 94% and 76%, respectively). After all, AL is NOT supposed to be for "quickly reaching mediocre performance," but rather "creating the best model with less data than required by random sampling." [...]”
> - We agree with the notion that AL is about "creating the best model with less data than required by random sampling”. Also, we agree that it is not about quickly achieving mediocre performance.
> However, we would like to point out that our performances on CIFAR-X of 94% (of ~96%) and 76% (of ~79%) are very close to SOTA performance with a ResNet-18 on the whole dataset annotated (in brackets). The accuracies you mention are essentially always achieved with very large models pretrained on ImageNet21k/1k [a]. Using methods like this would greatly improve the computational expense for running Active Learning experiments and, in the case of CIFAR-X lead to results that are also favorably biased as the classes are a subset of ImageNet that do not generalize to datasets/tasks where this is not the case.
> However, we agree that finetuning of foundation models has the potential to improve the baseline performance of all models greatly and added this to our conclusion & take-aways section.
>
>
> Thank you again for your valuable feedback. As we believe to have resolved your comments, please let us know in case there are remaining concerns that would prevent you from recommending stronger acceptance of our work.
>
> *Additional references*:
>
> [a]Foret, Pierre, et al. "Sharpness-aware minimization for efficiently improving generalization." arXiv preprint arXiv:2010.01412 (2020).

---

### Official Review · Reviewer_DMVM · 2023-07-06

**Soundness:** 3 good
**Presentation:** 2 fair
**Contribution:** 2 fair
**Rating:** 5
**Confidence:** 4

**Summary:**

The submission presents a large collection of experimental results for four active learning methods (entropy, core-set, BALD, BATCH) applied to image classification using ResNet-18 models, investigating the effect of differences in the data distribution (e.g., "roll-out data" from a different domain), size of initial labeled dataset, size of query batch, and integration of self- and semi-supervised learning. Special care is taken to optimize the supervised baseline and the active learning methods (on the development datasets).

**Strengths:**

The submission presents an extensive collection of results and appears to be the only study so far that jointly considers all the five "pitfalls" referred to in the manuscript.

The poor performance of semi-supervised learning on the roll-out datasets is remarkable, and the results bolster the case for the usefulness of active learning.

It is very useful to see an independent evaluation of BADGE that indicates its performance is robust.


**Weaknesses:**

Two of the main findings do not appear to be novel: the usefulness of careful parameter tuning and the beneficial effect of self-supervised learning (although this new work perhaps provides more extensive empirical support).

The results are based on a single neural network architecture (ResNet18) and a single performance metric on each of the two groups of datasets (plain accuracy on the development datasets and balanced accuracy on the roll-out datasets).

Only a single semi-supervised learning algorithm is considered (a version of FixMatch). Considering the statement that "Semi-SL performs best on toy datasets but does not seem to generalize to more complex settings.", this seems problematic.

The experiments seem to jointly vary the batch size and the size of the initial labeled dataset. It seems difficult to disentangle the effect of these two parameters based on the presentation of the results in the paper, and it is unclear whether all the claims regarding the effect of these two parameters are supported.

The submission implies that active learning is particularly useful when dealing with imbalanced data. However, it does not appear to consider techniques applied in standard supervised learning to deal with imbalanced data.

Full results for semi-supervised learning on MIO-TCD and ISIC-2019 should be included.

-----
The rebuttal has satisfactorily addressed these comments.
-----

**Questions:**

N/A

---

> ### Author Rebuttal · Authors · 2023-08-09
>
> We sincerely thank you for taking the time to carefully review our paper as well as for your insightful comments and questions. Note that the major changes made in our manuscript are detailed in the general reply above.
>
> -> “Two of the main findings do not appear to be novel: the usefulness of careful parameter tuning and the beneficial effect of self-supervised learning [...].”
> - We believe there is a misconception about the main concepts of our work. We have revised the description of these concepts (see general reply): AL validation paradox, P1-P5, and roll-out data sets. We hope this update resolves the misconception. Essentially our argument is that all five pitfalls need to be avoided in order to be able to make meaningful statements about the practical value of QMs. As shown in Table 1, *none of the related work currently checks all boxes*. Thus, even though some of our findings are not entirely novel, our holistic approach enables statements about the practical value of AL methods such that they transfer to a wide range of application scenarios. Further, the approach implied in your comment about hyperparameter tuning using AutoML on every AL training step [39] is substantially more expensive (factor of 10) compared to our light-weight protocol, where hyperparameters are tuned once on the starting budget without compromising classifier performance (see Appendix H). Our contribution therefore includes a practically feasible solution that might enable widespread adoption in the field.
> Similarly, regarding the efficiency of self-supervised learning, we are the first to verify the advantage over well configured supervised baselines on multiple and imbalanced datasets.
>
> -> “The results are based on a single neural network architecture [...] and a single performance metric on each of the two groups of datasets [...].”
> - The most widely used architectures in the deep AL classification community are VGG and ResNet18 [2,4,6,25,30,39,43,45,49,50,53]. We focused on the ResNet-18 as it performs better and allows for simple integration of Self-SL and Semi-SL paradigms. Please also note that other reviewers commented on our experiments as “thorough” and “extensive”.
> - Deriving insights based on a single classification metric per dataset is the standard in AL literature [4,6,11,18,19,25,38,39,49,53]. Exceeding the usual analysis, we added two metrics specific to AL evaluation. The Area Under Budget Curve[52] and the Pairwise Penalty Matrix [2]. Nevertheless, to resolve your concern regarding the small number of employed metrics, we added results to Appendix F and the PDF alongside the general reply showing macro-averaged F1-scores for ISIC-2019 and MIO-TCD.
>
>
> -> Problematic statement about Semi-SL:
> - Thank you for pointing this out. We agree that our original statement about Semi-SL methods was too broad. As detailed in the general reply, we toned down our statement.
>
> -> “The experiments seem to jointly vary the batch size and the size of the initial labeled dataset. It seems difficult to disentangle the effect of these two parameters based on the presentation of the results in the paper, and it is unclear whether all the claims regarding the effect of these two parameters are supported.”
> - We correlated the two parameters (smaller query size for smaller starting budget etc.) since 1) in practice, we deem the size of the starting budget to be dependent on labeling cost (therefore, large query sizes for small starting budgets are unrealistic and vice versa) and 2) In this work, we are especially interested in smaller starting budgets (“cold-start” territory) compared to the ones in the literature, since AL typically shows robust performance for larger starting budgets. Theory shows that our adapted smaller query size for this case can only positively affect the result [27]. The only possible confounder could be that we interpret the performance of a small starting budget too positively due to a hidden effect of the smaller query size. However, we performed an ablation study in Figure 3, showcasing that varying the query size for small starting budgets did not have considerable effects on performance for all QMs, except BALD, where expectedly, a clear performance increase for smaller query sizes was observed.
> Thank you for pointing out a lack of clarity in this design choice. We thoroughly revised the respective parts in our manuscript.
>
> -> “The submission implies that active learning is particularly useful when dealing with imbalanced data. However, it does not appear to consider techniques applied in standard supervised learning to deal with imbalanced data.”
> - As stated in Section 3 (lines 250-254), we apply balanced sampling of classes for standard training and Self-SL (finetuning) and weighted CE-Loss for Semi-SL. These are standard techniques to deal with imbalanced data. To further resolve your concern, we now provide an additional ablation for all experiments using weighted-CE-Lossi for ST and Self-SL models. For results, we kindly refer to the general reply and the provided PDF.
>
> -> “Full results for semi-supervised learning on MIO-TCD and ISIC-2019 should be included.”
> - Expectably, on CIFAR-X datasets we observed the gain of Semi-SL over ST to be decreasing with more annotated data (i.e. largest gain on the starting budget). Since on MIO-TCD and ISIC-2019, we observe no Semi-SL gain already on the starting budgets of the low- and mid-label regime, it is reasonable to assume that no gain (over ST and Self-SL) will occur with more annotated data. Given this reasoning and that adding these experiments requires approximately 18k GPU hours, we think it is fair to omit the experiments. We clarified this reasoning in the manuscript.
>
> Thank you again for your valuable feedback. As we believe to have resolved your comments, please let us know in case there are remaining concerns that would prevent you from recommending acceptance of our work.
>
> We use the citation keys from our manuscript.

---

> > ### Comment · Reviewer_DMVM · 2023-08-13
> > **Response to rebuttal**
> >
> > Thank you for your thoughtful responses. I will upgrade my rating for your paper accordingly.

---

> > > ### Author Response · Authors · 2023-08-21
> > >
> > > Thank you very much for taking the time to read our response and re-considering your assessment based on the provided updates.

---

### Official Review · Reviewer_WUPX · 2023-07-07

**Soundness:** 4 excellent
**Presentation:** 4 excellent
**Contribution:** 4 excellent
**Rating:** 7
**Confidence:** 4

**Summary:**

Active Learning is a crucial problem that focuses on selecting a subset of examples from an unlabeled dataset to be labeled. Primarily community focus on two pillars for defining acquisition functions -- uncertainty and diversity based acquisition methods. With emergence of the highly powerful pretrained models and alternative paradigm such as self SL and semi SL, the usecase of when to use AL is getting questionable. That being said, this work begins with listing down pillars at which AL scheme should be evaluated, starting from the hyperparameters to proper comparison against SSL baselines. Paper list down total 5 "pitfalls" and show how existing community has not been evaluating the proposed AL methods correctly. Paper then extensively evaluate the common AL QMs on different settings, and then giving concluding remarks on some of the existing methods, and guidance on when is AL beneficial.

**Strengths:**

The paper is well written, easy to follow and targets a well motivated problem. Previous AL works often become cynical, and when not carefully studied the mentioned pitfall, it can give a false sense of label efficacy. Conducted experiments are solid, and the community would benefit with it.

**Weaknesses:**

Weakness are more of some additional experimental suggestions. Given the abundance of pretrained CLIP models, I think adding them would also have been a great idea. Secondly, in terms of difficulty, paper could've added some large scale datasets such as Imagenet variants. Lastly, one additional pillar could also be studying the OOD robustness. This was recently studied in the AL paper -- "Continual Active Learning" which aimed to speedup the active learning, and proposed additional evaluation based on robustness.

Lastly, there is a typo on Fig 2a, HL, where 10000 is written as 1000.


-  Das et al. 2022 Continual Active Learning. https://openreview.net/forum?id=GC5MsCxrU- / https://arxiv.org/abs/2305.06408

**Questions:**

Please refer to weakness section.

**Limitations:**

Please refer to weakness section.

---

> ### Author Rebuttal · Authors · 2023-08-09
>
> We sincerely thank you for taking the time to carefully review our paper as well as for your insightful comments and questions. Note that the major changes made in our manuscript are detailed in the general reply above.
>
> -> “Weakness are more of some additional experimental suggestions. Given the abundance of pretrained CLIP models, I think adding them would also have been a great idea. “
> - We agree that foundation models such as CLIP pretrained models hold great promise for Active Learning. However, given the already extensive scope of our work, we leave the necessary experiments to future work. As the focus of our work also lies in establishing proper evaluation protocols, these methods can be evaluated following our protocol. Based on your suggestion, we extended our conclusion and take-aways section with reference to CLIP and foundation models.
>
> -> “Secondly, in terms of difficulty, paper could've added some large scale datasets such as Imagenet variants.“
> - MIO-TCD Dataset is a large scale dataset with approx 500,000 images. Further, both ISIC-2019 and MIO-TCD are datasets that are commonly argued to represent real-world use-cases and should not be lumped together with simple datasets like CIFAR. To our knowledge, our work is one of the most extensive AL studies for deep learning based image classification. Other reviewers have commented on our experiments as “extensive” and “thorough” and the number, complexity, and diversity of our utilized datasets and settings (label-regimes, training strategies) already exceeds the standard in the field.
>
> -> “Lastly, one additional pillar could also be studying the OOD robustness. This was recently studied in the AL paper -- "Continual Active Learning" [a] which aimed to speedup the active learning, and proposed additional evaluation based on robustness.”
> - We agree that studying other characteristics of the model, such as OOD robustness, would allow for further characterization of AL methods. However, in our study, we aimed to improve the evaluation of the standard classification metric by which all AL methods are evaluated. To our understanding, [a] mainly evaluated OOD robustness as they suspect that continually trained models exhibit worse generalization performance against shifts similar to models pre-trained on a different source dataset [b]. Since our trainings do not employ continual learning, this particular problem does not directly apply to our benchmark. However, as the general notion of evaluating robustness (such as in [a]) is very valuable for the community, we added a sentence in the conclusion & take-aways section stating that testing the robustness of AL, especially in the light of novel training schemes like continual learning is a potential future extension of our protocol.
>
> -> Lastly, there is a typo on Fig 2a, HL, where 10000 is written as 1000.
> - We thank the reviewer for pointing this out. We changed the last tick in the x-axis from 1000 to 10000 in Figure 2 (CIFAR-10) high-label regime.
>
>
> Thank you again for your valuable feedback. As we believe to have resolved your comments, please let us know in case there are remaining concerns that would prevent you from recommending stronger acceptance of our work.
>
> *Additional References*:
>
> [a] Das et al. 2022 Continual Active Learning.
> https://openreview.net/forum?id=GC5MsCxrU- / https://arxiv.org/abs/2305.06408
>
> [b] Ash, Jordan, and Ryan P. Adams. "On warm-starting neural network training." Advances in neural information processing systems 33 (2020): 3884-3894.

---

> > ### Comment · Reviewer_WUPX · 2023-08-17
> > **Thanks for the rebuttal**
> >
> > I'd like to thank the authors for the rebuttal; However, I'd like to retain my rating.

---

> > > ### Author Response · Authors · 2023-08-21
> > >
> > > Thank you very much for taking the time to read our response.

---

### Official Review · Reviewer_GZfx · 2023-07-11

**Soundness:** 3 good
**Presentation:** 2 fair
**Contribution:** 2 fair
**Rating:** 5
**Confidence:** 2

**Summary:**

The authors in this work try to address the common issues as to why active learning has not been in use compared to other methods like semi-supervised or self-supervised learning.  The authors argue that the active learning methods are not evaluated in robust manner and in a cross-task generalisation setting which leads to lower performance compared to semi-supervised and supervised learning. Hence, they also provide an evaluation framework to overcome these issues. The code for all their experiments is also made available.

This work is useful to anyone who is new to the field of active learning and are deciding whether or not active learning would be useful for their task and dataset.

The authors list five common pitfalls in the evaluation of active learning settings and suggest means to address these issues. Some of these issues are lack of evaluated data distribution settings, lack of evaluated starting budget, lack of evaluated query sizes, neglection of the classifier configuration and neglection of alternative training paradigms.

Some conclusions of this work, like active learning could provide substantial gains in class-imbalanced settings, reiterating that BADGE is the most robust QM etc can help the readers intending to use AL in their works.

**Strengths:**

The authors conduct a thorough ablation study to evaluate the active learning gains by changing the parameters of only one of these issues and keeping the rest constant. This has helped in filling the research gap between theoretical and practical works to use smaller query sizes as a potential solution for the instabilities of certain QMs like BALD. It has also helped in recalibrating the potential gains from active learning measured by previous works which could be attributed to hyperparameter settings. By introducing a practical way to obtain the best hyperparamter settings, this work shows the actual gains of active learning without overstating it.

**Weaknesses:**

Since it was mentioned in the limitation that the training may not have been optimal for semi-SL method, how can the conclusion that "semi-SL performs best on toy datasets but does not seem to generalize to more complex settings" be made?

Presentation -
- Some lines contain too many pieces of information in a single line which can be organised in a more concise and clearer manner
- A breif description of the citation would help in following the work better. For ex, writing a few words on what [6] is in line 313 would make the presentation better
- The abbreviation ST was used in line 186 before clarifying the meaning in line 235 which can be confusing

**Questions:**

- Can you clarify the relation between the starting budget and the labelling regimes(high-label,low-label etc)
- Model selection for ST and Self-SL models is based on the best validation set epoch, while for Semi-SL models the final checkpoint is used. - Why?
- Do the cost for the checks of the generalizability of AL in real-world settings itself negate the purpose of AL?

**Limitations:**

The authors have clearly mentioned the limitations of their studies.

---

> ### Author Rebuttal · Authors · 2023-08-09
>
> We sincerely thank you for taking the time to carefully review our paper as well as for your insightful comments and questions. Note that the major changes made in our manuscript are detailed in the general reply above.
>
> -> “Since it was mentioned in the limitation that the training may not have been optimal for semi-SL method, how can the conclusion that "semi-SL performs best on toy datasets but does not seem to generalize to more complex settings" be made?””
> - Thank you for pointing this out. We agree that based on our setting, this original conclusion about Semi-SL methods was too broad. As detailed in the general reply, we toned down our statement and put it into the context of similar reports in related work.
>
> -> Issues with Presentation:
> - We thank you very much for pointing out these flaws and revised our work thoroughly to resolve them. Most notably:
>     - We decreased the information density in our manuscript by pushing less-relevant information to the Appendix. For instance, we removed section 2.3 to a new Section in the Appendix and instead provide a more intuitive and crisp explanation of the importance of our five pitfalls and other important concepts as described in the general reply.
>     - All references are given adequate descriptions to improve readability.
>     - As suggested, we introduce the abbreviation Standard Training (ST) in line 186 now.
>
> -> “Can you clarify the relation between the starting budget and the labelling regimes(high-label,low-label etc)”
> - We define label regimes as specific settings for the sizes of starting budget and query size. The starting budgets for the low-, mid- and high-label regime vary based on the number of classes (5x, 25x, 100x #Classes) with the exception of CIFAR-100 (5x, 10x, 50x #Classes) due to the different ratio of #Classes to the dataset size. The query size is for each label regime identical to the starting budget (assuming both sizes derive from the labeling cost of a given problem). By running AL experiments on these different settings (label regimes) we wanted to test how well QMs generalize given different sizes of starting budgets. We were especially interested in the low- and mid-label regimes to see how well QMs perform with starting budgets which are usually attributed to be prone to the cold-start problem but one might unknowingly run into unto application. On the entanglement of starting budget and query size: The theory shows that our adapted smaller query size for this case can only positively affect the result compared to a larger query size [27]. Therefore, we could interpret the performance of a small starting budget too positively due to a hidden effect of the smaller query size. However, we performed an ablation study in Figure 3, showcasing that varying the query size for small starting budgets did not have considerable effects on performance for all QMs, except BALD, where expectedly, a clear performance increase for smaller query sizes was observed. We thank the reviewer for pointing out the vagueness and clarified the corresponding Section 3 in the manuscript.
>
> -> “Model selection for ST and Self-SL models is based on the best validation set epoch, while for Semi-SL models the final checkpoint is used. - Why?”
> - For ST it is considered good practice to select the best checkpoint [39] and it is intuitive to follow the same approach for Self-SL (finetuning). However, for Semi-SL, the current practice is to take the last checkpoint since it is assumed that the performance generally improves over the span of the long training with little risk of overfitting thanks to including the unlabeled data pool in the training [46], whereas both ST and Self-SL (finetuning) are prone to overfitting in this setting.
> We argue it is important to follow the standard practice in these choices to ensure comparability of our results as well as adequate settings for all training strategies.
>
> -> “Do the cost for the checks of the generalizability of AL in real-world settings itself negate the purpose of AL?”
> - Thank you for pointing out this trade-off.
> Our message is: if done right, users can save substantial costs with AL. While our protocol increases computational efforts for the initial validation of a newly proposed QM, this effort has to be made only once and does not come with additional labeling cost. However, if validation of QMs is done with our protocol, it enables reliable performance estimates for QMs in diverse new settings and applications, such that users can make high-quality decisions on when and how to apply AL on their customized problems. Thus, our protocol enables users to benefit from AL upon application reliably. We added a respective sentence to the discussion in our manuscript.
>
> Thank you again for your valuable feedback. As we believe to have resolved your comments, please let us know in case there are remaining concerns that would prevent you from recommending acceptance of our work.
>
> We use the same citation keys as in our manuscript.

---

> > ### Comment · Reviewer_GZfx · 2023-08-17
> >
> > Thank you for the clarifications and the necessary updates to the paper. I will update my ratings.

---

> > > ### Author Response · Authors · 2023-08-21
> > >
> > > Thank you very much for taking the time to read our response and re-considering your assessment based on the provided updates.

---

### Official Review · Reviewer_QpNG · 2023-07-27

**Soundness:** 3 good
**Presentation:** 3 good
**Contribution:** 3 good
**Rating:** 6
**Confidence:** 4

**Summary:**

The paper identifies the pitfalls in current evaluation frameworks for AL, and proposed solutions that lead to a new evaluation framework (codebase) to tackle these problems. Empirical results reveal insights on the current state of AL methods.

**Strengths:**

- The paper aims to tackle a very timely and important topic on how to more systematically and realistically evaluate AL methods.
- The paper identifies several pitfalls in current AL evaluation and proposed solutions to overcome them.
- The paper presents quite thorough experimental results accompanied by code releasing for future work to build on.
- The key takeaways provide insights on the current state of AL literature.

**Weaknesses:**

- Line 71 AL validation paradox: It is not very clear to me why AL parameters cannot be adapted to the dataset at hand. Why can't one adapt or even change query method in an online fashion like in [1]? I agree that one should not "use the performance" in a hindsight to choose the best AL method but that doesn't imply that we are only allowed to use fixed "pre-configured AL settings" for all new dataset?
- While AL validation paradox appears to be the most important pitfall in AL evaluation, it's not immediately clear to me as how P1-5 addresses the validation paradox. Can the authors discuss the connection between AL paradox to P1-5?
- While the paper has emphasized the importance of testing AL methods on new datasets and on realistic settings, the datasets included in the current experiments are limited in number, modality (only images) and relatively less realistic (e.g., CIFAR). Results/takeaways can be more convincing if more realistic datasets are used, and even include modalities other than image.

[1] Active Learning by Learning. Hsu et al. 2015.

**Questions:**

See above.

**Limitations:**

The paper discusses its limitation in its experimental setup.

---

> ### Author Rebuttal · Authors · 2023-08-09
>
> We sincerely thank you for taking the time to carefully review our paper as well as for your insightful comments and questions. Note that the major changes made in our manuscript are detailed in the general reply above.
>
> -> “Line 71 AL validation paradox: It is not very clear to me why AL parameters cannot be adapted to the dataset at hand. Why can't one adapt or even change query method in an online fashion like in [a]? I agree that one should not "use the performance" in a hindsight to choose the best AL method but that doesn't imply that we are only allowed to use fixed "pre-configured AL settings" for all new dataset?”
> - We hope the updated description of the validation paradox, the pitfalls and the roll-out datasets (see general reply above) answer this question.
> The bottom line is: The validation paradox does indeed force us to, on a new unlabelled dataset, choose the best query method (QM) based on prior knowledge (note that this is in stark contrast to standard ML, where the optimal configuration on a new dataset can be found using a validation split). This is why the avoidance of P1-P5, and thus the thorough validation proposed in this work, is crucial to enable an informed choice in a new setting.
> As for your suggestion to use a live-adapting QM [a], this is ofcourse possible, but the reliability of this live-adaptation needs to be validated equally beforehand under different settings. As such, the adaptation rules itself can be seen as a “preconfigured AL setting” whose reliability needs to be checked before applied on arbitrary tasks.
>
> -> “While AL validation paradox appears to be the most important pitfall in AL evaluation, it's not immediately clear to me as how P1-5 addresses the validation paradox. Can the authors discuss the connection between AL paradox to P1-5?”
> - Thank you for pointing out the lack of clarity regarding these main concepts in our manuscript. We thoroughly revised the description (see general reply) and hope the concepts are clearly conveyed in the updated text.
>
> -> “While the paper has emphasized the importance of testing AL methods on new datasets and on realistic settings, the datasets included in the current experiments are limited in number, modality (only images) and relatively less realistic (e.g., CIFAR). Results/takeaways can be more convincing if more realistic datasets are used, and even include modalities other than image.”
> - We argue that the number and diversity of datasets and settings in this study do indeed provide sufficient empirical evidence for the insights we provide. For instance, both ISIC-2019 and MIO-TCD are datasets that are commonly argued to represent real-world use-cases and should not be lumped together with simple datasets like CIFAR. To our knowledge, our work is one of the most extensive AL studies for deep learning based image classification. Other reviewers have commented on our experiments as “extensive” and “thorough”. The number, complexity, and diversity of our utilized datasets and settings (label-regimes, training strategies) exceeds by far the standard in the field. Note that adding more datasets is computationally more complex for AL compared to standard ML tasks.
> Regarding the suggestion to add more modalities, we clarified in the manuscript that the scope of our work is deep active classification of images.
>
>
> Thank you again for your valuable feedback. As we believe in having resolved your comments, please let us know in case there are remaining concerns that would prevent you from recommending  a stronger acceptance of our work.
>
> *Additional references:*
>
> [a] Active Learning by Learning. Hsu et al. 2015.

---

> ### Comment · Reviewer_QpNG · 2023-08-14
>
> Thank you to the authors for the response. I will maintain my original score.

---

> > ### Author Response · Authors · 2023-08-21
> >
> > Thank you very much for taking the time to read our response.

---

### Author Rebuttal · Authors · 2023-08-09

We sincerely thank all reviewers for their valuable comments. While the reviewers generally agreed on the added value of our work (“timely problem”,“extensive evaluation”, “useful insights”),  concerns were raised regarding the clarity of the manuscript and the toning regarding Semi-SL. We have thoroughly revised our manuscript to clarify the respective parts and resolve these concerns.

Next to the point-by-point responses below, we address the main recurring comments here:

**Presentation of the core concepts**: The original presentation of the core concepts of our work seems to be the driver behind most of the raised questions and comments. Thus, we  updated the explanation of the “AL validation paradox”, and how it interconnects with the five pitfalls P1-P5 as well as the concept of “roll-out” datasets:

“When applying AL on a new, mostly unlabelled data set, one can not directly validate whether QM1 or QM2 performs better, as this would require labeling individual AL trajectories through the data set for each QM. This excessive labeling would directly contradict the goal of AL to reduce labeling (“validation paradox”). Notably, this is in contrast to standard ML, where a fixed labeled validation split suffices to optimize the model configuration. Thus, the validation paradox forces one to instead estimate how well certain QMs will perform on the given task based only on prior knowledge. The quality of this estimate directly depends on the prior knowledge available, i.e. on how extensively the respective QM has been validated in the first place. The application at hand typically comes with a given class imbalance and constraints on the size of the starting budget and the query size. To ensure that the prior knowledge about the performance of a QM covers such specific application constraints, the QM needs to be validated on a realistic range of all of these parameters. Not doing so causes P1-P3 and limits the reliability of the required performance estimates of a QM in practice. Further, in P4 (“sub-optimal classifier configuration”) and P5 (“lack of alternative training paradigms SSL/Semi-SL”) we argue that AL is costly and thus assessing the practical value of a QM must include comparison against cheaper alternatives. For instance, if the gain of a QM over random sampling can also be achieved by simply tuning the learning rate of the classifier, the QM provides no practical value (P4), and similarly, if the gain can be achieved by a single SSL pretraining, the QM provides no practical value either (P5). Finally, several hyperparameters have to be configured in AL application like the classifier configuration (P4), but also possible hyperparameters of the QM itself. Since it is not feasible to validate a QM for all settings in this high-dimensional parameter space, as required by the validation paradox, we instead propose to test the robustness of a few optimized settings on additional datasets (“roll-out datasets”) that were not part of the optimization process (“development datasets”).
Taken together, statements about the practical value of a QM require addressing all five pitfalls P1-P5 and testing on “roll-out datasets”. As Table 1 shows impressively, the proposed protocol is needed to close the gap between current AL literature and a meaningful assessment of QMs.”
Semi-SL verdict: We toned down our original verdict about SemiSL (“Semi-SL performs best on toy datasets but does not seem to generalize to more complex settings”) and added more supporting literature for this observation: ”Our results based on FixMatch indicate that Semi-SL methods perform well on datasets where they have been developed, but may struggle to generalize to more realistic scenarios such as class imbalanced data. This observation is in line with various findings in literature including Semi-SL methods other than FixMatch [a,b,c,4,7,20]”.

**Organization of the manuscript**: To improve the readability and clarity of the manuscript, we implemented structural changes as suggested by reviewer D3At:
The current section 2.3 has been moved to the appendix and has been replaced with a more concise and clear description of the core concepts (see statement above), including an intuitive motivation for the roll-out datasets.
Instead, we increase the size of Figure 2 making the main results more readable. We further point to Appendix F now, containing detailed and zoomed-in graphs of all experiments.

**Additional experiments**: Some reviewers suggested adding new datasets, modalities or network architectures.While we agree with the scientific interest in these experiments we argue that the empirical support for our insights already exceeds the standard in the field.  To our knowledge this work is one of the most extensive AL studies out there with most reviewers acknowledging our “thorough” and “extensive” experiments. Nevertheless, we tried to include as many suggestions as possible and now provide:
An ablation of our over-sampling strategy on imbalance data, using weighted CE loss (as suggested by reviewer DMVM). Showing 1) a performance decrease of weighted CE compared to over-sampling for ST, and 2) even though the strategy for handling imbalanced data influences the performance of AL methods, the observed gain of AL methods over random sampling seems to be consistent.
Macro-averaged F1-score as an additional performance metric for the imbalance datasets MIO-TCD and ISIC-2019 (as suggested by reviewer DMVM).  Showing that the general trend observed with balanced accuracy holds for the macro-averaged F1-score.


*Additional references*:

[a] Guo et al. “Robust Deep Semi-Supervised Learning A Brief Introduction”. 2022

[b] Guo et al. “Safe Deep Semi-Supervised Learning for Unseen-Class Unlabeled Data”. 2020

[c] Kim et al. “Distribution Aligning Refinery of Pseudo-label for Imbalanced Semi-supervised Learning”. 2020

---

### Decision · Program_Chairs · 2023-09-21

**Decision:**

Accept (poster)

**Comment:**

This paper tackles the challenges in evaluating Active Learning (AL) methods, proposing an evaluation framework and conducting extensive experiments. The reviewers agree that this is a timely topic and the pitfalls identified can be useful to the community. Some reviewers point out the room for improvement in terms of dataset diversity, deeper study on the hyperparameter effects, and more careful study of advanced imbalanced learning techniques that are being coupled with AL. The authors' rebuttal addresses concerns and clarifies key issues. The authors should take the reviewers' comments into account in the camera-ready version.